# NCP: Neural Correspondence Prior for Effective Unsupervised Shape Matching

**Souhaib Attaiki**
LIX, École Polytechnique, IP Paris
attaiki@lix.polytechnique.fr

**Maks Ovsjanikov**
LIX, École Polytechnique, IP Paris
maks@lix.polytechnique.fr

## Abstract

We present Neural Correspondence Prior (NCP), a new paradigm for computing correspondences between 3D shapes. Our approach is fully unsupervised and can lead to high-quality correspondences even in challenging cases such as sparse point clouds or non-isometric meshes, where current methods fail. Our first key observation is that, in line with neural priors observed in other domains, recent network architectures on 3D data, *even without training*, tend to produce pointwise features that induce plausible maps between rigid or non-rigid shapes. Secondly, we show that given a noisy map as input, training a feature extraction network with the input map as supervision tends to remove artifacts from the input and can act as a powerful correspondence denoising mechanism, both between individual pairs and within a collection. With these observations in hand, we propose a two-stage unsupervised paradigm for shape matching by (i) performing unsupervised training by adapting an existing approach to obtain an initial set of noisy matches, and (ii) using these matches to train a network in a supervised manner. We demonstrate that this approach significantly improves the accuracy of the maps, especially when trained within a collection. We show that NCP is data-efficient, fast, and achieves state-of-the-art results on many tasks. Our code can be found online: https://github.com/pvnieo/NCP.

## 1 Introduction

Establishing dense correspondences between 3D shapes is a fundamental problem in computer vision and computer graphics, as it enables many downstream applications such as statistical shape analysis [1, 2], texture [3] and deformation transfer [4, 5], and registration [6], to name a few.

A particularly challenging setting for this task is the computation of correspondences between 3D shapes that undergo non-rigid, non-isometric deformations, and that may exhibit some partiality, such as missing semantic parts, and can be represented as sparse point clouds.

The standard approach is to formulate shape correspondence as a supervised learning problem, by relying on ground truth maps within large datasets of shape pairs. Given such ground truth, it is possible to train neural networks to either produce deformation fields [7], segmentation maps [8, 9, 10] or functional maps [11, 12, 13] between the input shapes. However, all such methods rely on the presence of labeled point-to-point correspondences, which are expensive to obtain, and are only available in a handful of cases.

At the same time, *unsupervised* techniques try to solve the matching problem by imposing structural properties on the maps, either in the intrinsic [14, 15, 16] or extrinsic domains [17, 18]. However, the priors used by these methods tend to be purely geometric (e.g., promoting near isometry or divergence-free deformation fields), and, as we demonstrate below, are not always applicable, especially in challenging non-isometric settings.

36th Conference on Neural Information Processing Systems (NeurIPS 2022).

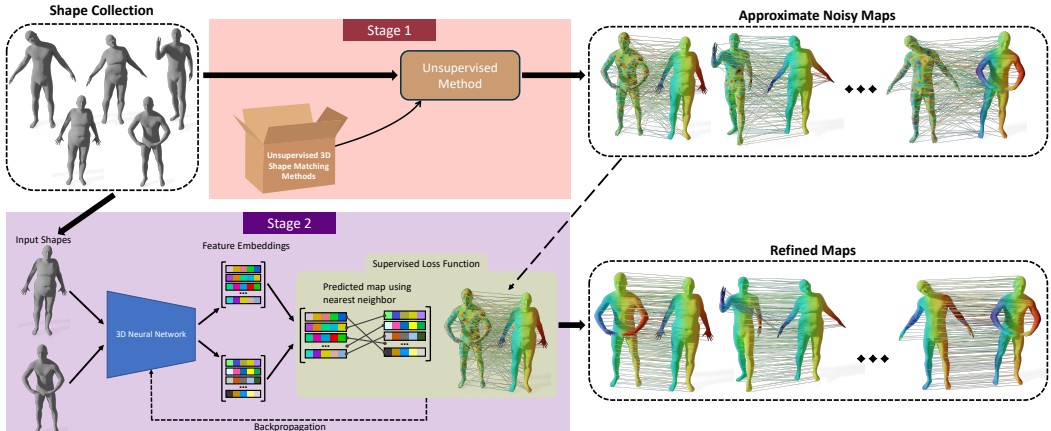

Figure 1: We present **NCP-UN**, an unsupervised shape matching method applicable even in challenging cases of non-isometric non-rigid shapes or sparse point clouds. Our method is composed of two stages (detailed in Algorithm 1). Namely, given a collection of shapes, we use an existing unsupervised method to obtain an initial set of potentially approximate, highly noisy maps. These maps are then used *as supervision* for a learning-based shape-matching method in Stage 2. Remarkably, we show that the resulting maps are of better quality than the maps used for training.

In this work, we demonstrate that the neural network itself can act as a powerful prior for shape correspondence problems. For this, we first observe that even without training, recent architectures for 3D shape analysis [19, 20, 21] produce pointwise features that capture local geometry and lead to well-structured maps between shapes. Secondly and perhaps more importantly, we demonstrate an effect that we call *neural correspondence prior*. Specifically, given noisy input maps, we formulate a supervised learning problem, where we aim to learn pointwise features that, when compared, would recover the input maps. Remarkably, we show that the correspondences computed by the trained networks are typically of higher quality than the input. Our results are consistent with the notion of *noise impedance* observed in other works on neural priors pioneered in [22], and *neural bias* [23] which states that neural networks tend to optimize lower frequency signals before higher-frequency ones. We show that in the context of shape matching, these effects imply that neural networks can learn powerful features by training to overfit to given noisy or incoherent correspondences.

Based on these observations, we develop a two-stage algorithm for unsupervised shape matching (see Fig. 1). In the first stage, we adapt an existing unsupervised neural network, using variants of standard methods from the literature. We then train a new network from scratch in a supervised manner using the noisy maps as ground truth. We demonstrate that this second stage not only helps to denoise the maps but also improves the matching result overall and achieves state-of-the-art results on multiple tasks. We show that *neural correspondence prior* is applicable within both collections and on individual shape pairs, to provide test time refinement of dense input maps. Since this method makes no assumptions about the geometry of the shapes, it can be used for complex cases such as non-isometric matching.

Overall, our contributions can be summarized as follows:

- We demonstrate that untrained neural networks for 3D shapes can produce pointwise features that are on par with complex axiomatic local descriptors.

- We show that when trained using artifact-laden maps for supervision, the features learned by neural networks lead to correspondences that are more coherent and of higher quality than the input. We call this effect *Neural Correspondence Prior*.

- Based on these insights, we develop a two-stage algorithm for unsupervised 3D non-rigid shape matching that achieves SOTA results for difficult matching scenarios such as non-isometric and point cloud matching.

- Using our unsupervised point-to-point correspondence method, we propose an approach for few-shot keypoint detection.

## 2 Related work

Shape matching is a well-studied area in computer vision and graphics, and a full overview is beyond the scope of this paper. We refer the interested readers to recent surveys [24, 25, 26, 27] for a more in-depth treatment of this field. Below, we review methods that are most related to our work, with a focus on learning-based techniques, both supervised and unsupervised.

**3D non-rigid shape correspondence**    There is a vast literature on learning non-rigid shape correspondences. In terms of supervised methods, an important direction is to consider the matching problem as a dense semantic segmentation problem where the labels are vertices on some template shape [8, 9, 10], or by mapping via template deformation [7]. However, these techniques are known to require a considerable amount of data, and may fail when discretization changes [19, 11]. Another approach is to use the functional map framework that was introduced in [28] and extended in many follow-up works [29, 30, 31, 32, 33, 34, 35, 36, 37] to name a few (see [38] for an overview). The functional map (fmap) framework is based on encoding and optimizing maps as small matrices in a reduced basis. This formulation has been successfully used in the supervised setting to establish correspondences between complete and partial non-rigid shapes [11, 12, 13]. However, it is typically restricted to near-isometric matching between relatively clean discretizations. The fmap framework has also been adapted to the unsupervised setting, either by imposing structural properties on the functional maps [15, 39], by enforcing cycle consistency [40], or by combining intrinsic and extrinsic properties [17, 18]. Another line of work proposes to perform unsupervised shape matching by reconstructing the input shapes in canonical order, based on an autoencoder [41, 42, 43]. However, these methods require a lot of shapes and learning time, and have only successfully been applied to man-made shapes that do not undergo significant non-rigid deformations.

**Neural prior**    The neural prior was first introduced in the seminal work of Ulyanov *et al.* [22], where the authors introduced the Deep Image Prior (DIP), and showed that a randomly initialized convolutional neural network can serve as a good prior for many 2D inverse problems, such as inpainting and denoising. Since then, several modifications and improvements have been made to DIP [44, 45, 46, 47, 48], in order to improve its performance or adapt it to new tasks. Using *untrained features* for matching was recently used in the context of 2D keypoint matching in [49]. Another line of work [50, 51] has theoretically studied DIP either through the lens of regularization theory or through its connection to Gaussian processes. Despite its success in 2D computer vision, neural prior has been little exploited in the 3D domain, the only application being shape reconstruction. Indeed, [52, 53] attempt to reconstruct a water-tight mesh from a point cloud, either by overfitting several local patches to the point cloud or by shrinking an input mesh by a neural network. The objective of our work is to fill this gap and to propose a neural prior technique for 3D matching.

**Unsupervised keypoint detection**    Unlike the 2D case [54, 55, 56, 57], unsupervised 3D keypoint detection is relatively under-explored in the literature. The recently introduced KeyPointNet [58] benchmark provides a good testing bed. Prominent unsupervised keypoint detection methods are based on a self-supervised paradigm using an encoder-decoder network, with a reconstruction error. The encoder of Skeleton Merger [59] predicts a set of salient keypoints, that the decoder uses to produce a skeleton of the shape. UKPGAN [60] harnesses the power of GANs to detect keypoints, by forcing the decoder to reconstruct the input shape using solely the set of keypoints produced by the encoder. Finally, KeypointDeformer [61] predicts a set of keypoints that allow for efficient shape manipulation. In particular, given a source and a target shape, the keypoints predicted on the source shape are used to deform it to the target shape using a cage skinning deformer [62].

## 3 Motivation & Method overview

Our main goal is to develop an unsupervised learning-based shape correspondence method that would take as input a pair of shapes $\mathcal{M}$ and $\mathcal{N}$ and produce a dense correspondence, also called a map, $T : \mathcal{M} \to \mathcal{N}$. For this, we explore the direction of neural priors and show how they can be used for 3D shape matching. The motivation behind this direction is twofold. First, as in the 2D case [63, 46], we hope that the structure of an untrained 3D neural network captures the low-level statistics of a single 3D shape, and thus can enable fine-tunning using a single pair. Second, as demonstrated in DIP [22], deep neural networks have high noise impedance, which allows them to learn good features from noisy inputs. This property can be used to guide the network to good local minima while trying to use, in our case, noisy point-to-point (p2p) matches for supervision.

| Methods | Method type | FAUST | SCAPE |
|---|---|---|---|
| HKS [69] + NN | *Axiom* | 26.8 | 28.6 |
| WKS [70] + NN | *Axiom* | 24.8 | 26.0 |
| SHOT [71] + NN | *Axiom* | 37.7 | 38.4 |
| DiffusionNet [19] + NN | *Rand.init* | $17.8 \pm 0.1$ | $20.3 \pm 0.3$ |
| BCICP [33] | *Axiom* | 15. | 16. |
| FMNet [13] | *Sup* | 11. | 17. |
| SURFMNet [15] | *Unsup* | 15. | 12. |
| Unsup FMNet [14] | *Unsup* | 10. | 16. |
| HKS + FMAP | *Axiom* | 20.2 | 26.9 |
| WKS + FMAP | *Axiom* | 20.1 | 27.1 |
| SHOT + FMAP | *Axiom* | 19.8 | 20.0 |
| DiffusionNet + FMAP | *Rand.init* | $8.9 \pm 0.02$ | $13.9 \pm 0.3$ |

*Right plot — NCE loss (—) vs. Geodesic error (---) over Optimization iterations. Legend:*
- map (a) - 0.0
- map (b) - 14.8
- map (c) - 28.5
- map (d) - 41.5

Figure 2: **Motivation for the NCP effect.** Left: Matching accuracy of DiffusionNet *with random weights* (mean and standard deviation of three random runs), compared to baseline shape matching methods, on the test sets of FAUST and SCAPE. Right: The evolution of the training NCE loss and geodesic error during optimization when using the ground truth map (map (a)), and noisy maps with different levels of noise (maps (b), (c) and (d)) as supervision. The legend reports the geodesic error of the input maps. Observe the noise impedance and geodesic error decrease of *intermediate* maps.

**Feature embeddings.** In this work, we formulate the shape-matching problem for both man-made and organic shapes, by learning pointwise features. Such features can be used to compute correspondences either through nearest neighbor search or with minimal post-processing. Specifically, the network $N$ takes as input a 3D shape $\mathcal{M}$ and produces, as output, a feature vector $N(\mathcal{M})_x \in \mathbb{R}^d$, for every point of the shape $x \in \mathcal{M}$. When considering the entire object as a whole, we call the set $N(\mathcal{M})_x$ for all $x \in \mathcal{M}$ a *feature embedding* of shape $\mathcal{M}$.

Our method is based on two main observations: *neural bias* for untrained networks, and an effect that we call *neural correspondence prior*. Below we describe each of these effects and then describe our unsupervised shape-matching method in Sec. 4.1.

## 3.1 Neural bias for pointwise features

Our first observation is that feature embeddings produced by neural networks have a particular structure, which can be exploited in the context of computing features for shape correspondence. To highlight this, we considered the test sets of the FAUST-Remeshed (FAUST) and SCAPE-Remeshed (SCAPE) datasets introduced in [33], and used in many recent works [11, 17, 39], and computed the correspondences using the pointwise features extracted by a DiffusionNet network [19] with randomly set weights. We use the default variant of DiffusionNet, which takes as input the XYZ coordinates of the shapes and produces a 128-dimensional feature vector for every point. We emphasize that the weights of the network are set randomly and *without any training*. We then used the extracted features to produce p2p maps either via a simple nearest neighbor (NN) search in the feature space or with the functional map framework [64] (FMAP). We compare the results to maps produced using the same procedure with classical axiomatic features such as SHOT [65], or using other axiomatic and training-based non-rigid correspondence methods.

As shown in Fig. 2-Left, remarkably, the features produced by an *untrained* DiffusionNet network perform on par or better than axiomatic features and even outperform the *supervised* learning method, FMNet [66], based on training MLPs to refine pre-defined SHOT features. We attribute the fact that a randomly-initialized DiffusionNet performs better than *trained* point-wise MLPs to the spatially-aware nature of the architecture of the network, which uses diffusion to simulate intrinsic convolution, thereby providing a strong neural prior [67, 68, 53].

## 3.2 NCP: Neural Correspondence Prior

While the previous effect relates to properties of untrained networks, in this work, we also observe another, complementary phenomenon that we call *Neural Correspondence Prior*, and which we exploit in our approach below. This effect can be formulated as follows. Suppose we are given a fixed map $T_{\mathcal{MN}}$, between some shapes $\mathcal{M}$ and $\mathcal{N}$. We can formulate an optimization problem, where we aim to learn the feature embedding of $\mathcal{M}$ and $\mathcal{N}$ so that the induced map, e.g., computed via nearest neighbors in the feature space: i.e. $x \rightarrow \arg\min_y \|N(\mathcal{M})_x - N(\mathcal{N})_y\|$, is as close as possible to $T_{\mathcal{MN}}$. Note that unlike the neural bias mentioned in the previous section, here we formulate an optimization problem, where the parameters of the network $N$ are optimized to fit the given input

map. Our key observation is that when the target map $T_{\mathcal{MN}}$ is noisy or approximate, networks tend to produce well-structured feature embeddings until the very late stages of the optimization. I.e., throughout the early stages of training, the maps computed by the optimized features tend to be of *higher accuracy than the input* noisy map used for supervision.

To demonstrate the effect of NCP quantitatively, we performed an experiment in which we corrupted the ground truth map between a pair of shapes from the FAUST dataset, and then trained the Diffu­sionNet network, with a gradient descent optimizer using the standard NCE loss [72] to overfit to those corrupted maps. Specifically, to supervise this training, we use: (*a*) the ground truth (GT) map between the shapes, (*b*) the GT map where 25% of its entries have been assigned to random vertices, (*c*) the GT map with 50% noise, (*d*) the GT map with 75% noise. The results of this experiment are shown in Fig. 2-Right. We plot both the NCE loss being optimized as well as the *geodesic error* of the intermediate maps with respect to the ground truth.

As shown in Fig. 2-Right, while the network can easily adapt to the correct map, it has difficulty minimizing the loss in the case of noisy maps. Moreover, and perhaps more remarkably, observe that the geodesic error starts to decrease *towards lower values than the input map*, in the first iterations of optimization and diverges only in the later stages, when the network starts to overfit to the noise in the training map. This implies that such optimization with a neural network and early stopping can be used as an effective map denoising mechanism (Sec. 4.2). It should be mentioned that this effect does not depend on the shape pair, and other examples of different pairs are provided in Sec. 5.1.2 and the supplementary.

We observe this effect to hold very broadly in shape correspondence problems. For a single shape pair with a noisy input map, eventually, any map can be learned by the network if the parametrization and the number of training iterations are sufficient. However, the network architecture strongly regularizes the map search space, providing low impedance to 'signal' and high impedance to 'noise', resulting in an optimization trajectory that either converges to a good local minimum or passes near one [22]. Furthermore, in the presence of *shape collections*, this effect is even stronger and, as we demonstrate below, significantly helps to regularize computed features so that *no early stopping* or post-processing of the results becomes necessary (see our Algorithm 1 that we explain in detail in Sec. 4.1).

In addition to the network architecture itself, we attribute the *Neural Correspondence Prior* to the spectral bias principle [23, 73], which states that neural networks tend to learn low frequencies in the early stages of training and that low frequencies are more robust to random perturbations of network parameters. Since we formulate shape matching as the problem of computing optimal features, spectral bias, in our context, implies that even given a noisy or artifact-laden map as input, the *feature embeddings* produced by the optimized neural network tend *to be smooth*, especially in the early stages of the optimization. In other words, it is significantly harder for the network to produce a noisy, high-frequency feature embedding than a smooth one. Furthermore, a smooth feature embedding will tend to produce smooth correspondences between shapes, as suggested by the following theorem, which we state here and prove in the supplementary.

**Theorem 1.** *Let $\mathcal{M}$ and $\mathcal{N}$ be two compact smooth surfaces (smooth manifolds of dimension 2). Let $\mathbf{M}$ and $\mathbf{N}$ be their feature embeddings in $\mathbb{R}^d$, given by some functions: $\psi : \mathcal{M} \to \mathbb{R}^d$ and $\phi : \mathcal{N} \to \mathbb{R}^d$, so that $\mathbf{M} = \psi(\mathcal{M})$ and $\mathbf{N} = \phi(\mathcal{N})$. For example, $\psi$ and $\phi$ can be given by some neural network. Suppose that $\psi$ and $\phi$ are both smooth and injective. Then up to arbitrarily small perturbations of $\phi, \psi$, the map $T_{nn} : \mathcal{M} \to \mathcal{N}$ given by $T_{nn}(x) = \arg\min_{y \in \mathcal{N}} \|\psi(x) - \phi(y)\|$ must be smooth up to sets of measure 0 on $\mathcal{M}$.*

This result highlights the fact that smooth feature embeddings, generically, translate into smooth maps, which is a desirable property in most typical correspondence scenarios.

# 4 Method description

Based on the observations above, in this section, we introduce contexts in which the NCP can be exploited, resulting in two algorithms: one for unsupervised shape matching, and the second for p2p map denoising.

## 4.1 NCP within a shape collection

Given a collection of shapes $\{\mathcal{N}_i\}$, we propose to exploit the NCP in two ways. The first configuration assumes that we are also given a collection of artifact-laden maps $\{T_j\}$ between some shape pairs in

**Algorithm 1:** NCP-UN: **N**eural-**C**orrespondence-**P**rior based **UN**supervised Shape Matching

**1**     **Input:** Collection of shapes $\{\mathcal{N}_i\}$

**2**     **Output:** p2p maps between shapes of the test set of $\{\mathcal{N}_i\}$

Stage 1 $\Big\{$
   1: Train a variant of an off-the-shelf unsupervised matching method **UM** on the train set of $\{\mathcal{N}_i\}$
   2: Predict p2p maps $\{T_i\}$ on the train set of $\{\mathcal{N}_i\}$ using trained **UM**

Stage 2 $\Big\{$
   3: Use $\{T_i\}$ to supervise the training of a randomly-initialized neural network **RN** with a p2p loss
   4: Predict Feature Embedding $\{\mathbf{RN}(\mathcal{N}_i)\}$ on the test set of $\{\mathcal{N}_i\}$ using the trained **RN**
   5: Use $\{\mathbf{RN}(\mathcal{N}_i)\}$ to establish correspondences either using the nearest neighbor or functional map pipeline.

$\{\mathcal{N}_i\}$ as input. In that case, we propose to train neural network to produce feature embedding for each shape, s.t., the induced maps are as close as possible to the target maps $\{T_j\}$. We provide the exact choice of the loss function depending on the nature of the underlying shapes in Sec. 5 below.

In the second configuration, no maps are given as input. In this case, we propose a two-stage pipeline where in the first stage, we adapt an existing unsupervised correspondence method to obtain possibly approximate very noisy initial correspondences. We then formulate a supervised learning problem, where we learn feature embeddings that would induce these computed correspondences as a target. Once the network is trained, we establish final correspondences via a simple nearest neighbor search in the feature space. Note that the trained network can be used to establish correspondences both between shapes within the given shape collection, but also *across new, never seen shape pairs.* In our evaluation below, and unless otherwise noted, we always evaluate on a test set, never seen during training. We dubbed the resulting algorithm **NCP-UN**, and we summarize it in Algorithm 1 and Fig. 1. This is the algorithm that is used throughout most of our experiments.

We argue that it is advantageous to perform Stage 2 simultaneously on an entire shape collection, instead of doing it for each pair individually for two main reasons: first, it is faster and avoids the need for test-time optimization (see Sec. 4.2), and second, the collection provides an additional layer of regularization. In fact, since we optimize for features on each shape, it is difficult to compute features that would overfit to all $O(N^2)$ noisy maps in a collection of $N$ shapes, as the artifacts across pairs are typically not consistent. We observe that the noise impedance property is even stronger in the case of a collection, and the produced maps are better than if the refinement is done per pair.

**Intuition behind Stage 2 of Algorithm 1**   Our intuition for Stage 2 is threefold. First, and motivated by the results of Sec. 3.2 (the NCP effect), we showed that neural networks tend to have difficulty overfitting to corrupted inputs, and tend to favor smooth outputs. Thus, in the process of over-fitting to approximate maps by a neural network, the output maps tend to be of **better quality** than the input. This motivates the use of artifact-laden maps as supervision for a network thus capitalizing on the NCP effect to recover better correspondences. Second, as we mentioned above, given a collection of approximate maps, their errors tend to be inconsistent, and training a network using such maps for supervision adds an extra layer of regularization, as it is difficult for the network to adapt to all inconsistent errors on all pairs at once. Finally, in our formulation of Stage 2, we learn *one feature embedding per shape* and compute correspondences via nearest neighbor search between feature embeddings. This also adds strong regularization, since it makes it difficult for the network to learn a feature embedding, which can reproduce errors in the maps between all shape pairs.

For our applications (see Sec. 5), we used the following design choices. For the **UM** method, we used the unsupervised functional map method [15, 39], with a DiffusionNet [19] backbone for 3D meshes, and Point-MLP [74] backbone for point clouds. The same backbone was used as **RN** for step 3. in Algorithm 1. Concerning the p2p loss in step 3, we used the LIE loss [75] for 3D meshes and point clouds. We also used a variant of our approach, dubbed NCP-UN$_{\text{fmap}}$, that uses the FMAP framework and loss from [11] in steps 3 and 5 of Algorithm 1, especially on organic shapes, where the Laplacian basis tends to be of high quality.

We want to emphasize that our observations regarding NCP are independent of the choice of loss and network architecture. A demonstration of the generality of NCP, as well as more details about the implementation, are provided in the supplementary. Across *all* cases, we observe a significant improvement in results when using the second stage of our NCP-UN pipeline (supervised learning) compared to the initial input maps.

Finally, we want to emphasize that it is possible to repeat Stage 2 several times, and we observed that there is a slight improvement in some cases, but most often it stagnates after the first iteration. We experimented with this and chose to keep the method simple and use only one iteration, thus, avoiding an additional tunable hyperparameter.

## 4.2 Test time denoising

In addition to the previous setting, we also observe that NCP can be used to denoise a correspondence when given a single shape pair. Specifically, given a noisy initial map, we train a neural network to produce feature embeddings that would induce the given input map, used as supervision. Again, we tailor the exact choice of loss to the nature of the shapes and describe one option in Sec. 5.1.2 below.

Test time denoising can be advantageous in the absence of a shape collection. At the same time, as mentioned in the original DIP work [22], in the case of a single shape pair, early stopping must be used in order to stop training at the best local minimum. For the shape-matching task, we use a cycle loss as the criterion for stopping. Given a pair of shapes $\mathcal{M}, \mathcal{N}$, and two maps between them $\Pi_{\mathcal{MN}}$ and $\Pi_{\mathcal{NM}}$ computed via nearest neighbors between optimized features, and represented as soft binary matrices (see Appendix A for computation details), we compute $L_{cycle} = \|X_{\mathcal{N}} - \Pi_{\mathcal{NM}} \Pi_{\mathcal{MN}} X_{\mathcal{N}}\|_F^2$, where $X_{\mathcal{N}}$ is the matrix of XYZ coordinates of shape $\mathcal{N}$. We overfit the network for a number of iterations and stop when $L_{cycle}$ stops improving after a predetermined patience period. The weights that produce the minimum value of $L_{cycle}$ are used to compute feature embeddings that we then use to compute the p2p map, via nearest neighbor search.

## 4.3 Correspondences as a tool for downstream tasks

Given computed correspondences, we also use them to solve multiple downstream tasks. Indeed, since our proposed algorithm is unsupervised, we can use its output, i.e., p2p maps, to transfer multiple quantities between shapes, such as keypoints. For this, we only need a few labeled examples with the quantities we want to transfer, resulting in a few-shot algorithm.

In the following, we present a method for few-shot keypoint detection, dubbed **FSKD** hereafter, and based on the p2p maps obtained by our unsupervised algorithm **NCP-UN**. A particular challenge in this setup is that some keypoints and parts appear exclusively on some shapes and not in others, and so to account for the keypoint appearing there, multiple labeled source shapes must be used.

We begin our **FSKD** by randomly selecting $N$ shapes that cover the largest set of keypoints. These $N$ shapes will be labeled and we transfer their keypoints to the shapes in the test set, in order to "discover" their keypoints.

Given a target shape, **FSKD** is composed of three steps: **1.** Detection of potential keypoints by transferring keypoints from labeled shapes, **2.** Filtering to remove keypoints that are likely not to exist on the target shape, **3.** Combination: merge transferred keypoints if multiple points on the target shape are assigned to the same keypoint ID. We provide the implementation details for these three steps in the supplementary (see Appendix C ).

# 5 Results & Applications

In this section, we provide results in a wide range of challenging tasks, showing the efficiency and robustness of our approach to different types of data and tasks. In particular, we consider the tasks of shape correspondence between man-made shapes in Sec. 5.1.1, non-rigid shape correspondence in Sec. 5.1.2, part segmentation in Sec. 5.2, and few-shot keypoint detection in Sec. 5.3. The fact that our model does not rely on a geometric prior allows it to provide good results on different types of data, including isometric and non-isometric, organic, and man-made data.

**Datasets** We conducted our experiments on four different datasets. To the best of our knowledge, there is no point cloud dataset that provides dense point-to-point ground-truth correspondences. For this purpose, we follow the setup of [41], and use KEYPOINTNET dataset [58] for the man-made shape correspondence experiment in Sec. 5.1.1. We also use this dataset for our few-shot keypoint detection experiment in Sec. 5.3. KEYPOINTNET is composed of over 8K models and 100K semantic keypoints labeled by professional humans. We use the same train/validation/test splits as the original paper [58]. For the part segmentation experiment in Sec. 5.2, we use the PARTNET dataset [76],

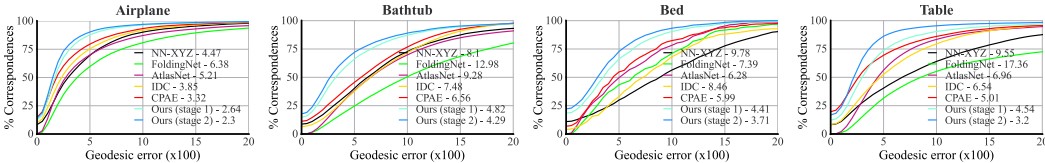

Figure 3: Correspondence accuracy on 4 categories in the KEYPOINTNET dataset.

which is composed of 16881 shapes of 16 categories, where each category has a different number of parts ranging from 2 to 6.

For 3D meshes, we use the organic, non-isometric non-rigid SMAL [77, 36] dataset and the SHREC'20 Non-Isometric benchmark [78] for our non-rigid correspondence experiment in Sec. 5.1.2, as these are challenging *non-isometric* datasets. The former dataset is composed of 50 animal shapes represented as 3D meshes. 25 shapes are used for training, and the remaining 25 are used for testing. It should be noted that the set of animals and poses seen in training is different from the one used in testing. The SHREC'20 benchmark consists of 14 scans of different animals, some of which contain holes and topological deformations. This, in addition to the fact that the ground truth maps are given as a set of sparsely annotated keypoints, means that most supervised methods cannot be applied.

## 5.1    3D shape correspondences

### 5.1.1    Correspondences on point clouds

We test the quality of the matches produced by NCP-UN on sparse point clouds using the KEYPOINT-NET dataset. We train the first stage of our method using the unsupervised functional map method [15, 39], with a PointMLP backbone [79] and XYZ coordinates as input, and impose the bijectivity loss on the functional map. The second stage is trained using LIE loss, and maps are extracted using the nearest neighbor in the feature space. For testing, we follow the same setup as [41], by generating pairs of shapes between all point clouds in a given category and removing the pairs that do not share the same set of semantic keypoints. We compute the L2 distance between the transferred keypoint and the ground truth one. Our method is evaluated against state-of-the-art learning-based 3D dense correspondence prediction approaches, including AtlasNetV2 [80], FoldingNet [81], IDC [43], CPAE [41], in addition to the nearest neighbor baseline between the XYZ coordinates of the points, which turns out to be a competitive baseline, not considered by previous methods.

In Fig. 3, we report the percentage of testing pairs where the distances between predicted and ground truth maps are below a given threshold for 4 categories and report the remaining categories in the supplementary. Our method outperforms all methods in 13 out of 16 categories. Note that our method obtains SOTA results although the method used in stage 1 is not fully designed to work on point clouds (the quality of the Laplacian is not good, and the used bijectivity loss is weak). We believe that using a more adapted method for the first stage will further improve the results.

### 5.1.2    Correspondences between non-rigid meshes

For the non-rigid non-isometric application, we trained the unsupervised functional map network from [19] and used LIE loss for the second stage (step 3. in Algorithm 1). P2P maps were obtained using nearest neighbors between feature embeddings. Because the SHREC'20 dataset is limited (only 14 shapes), we follow previous work [82] by training and testing on the same set, whereas for SMAL, the training and testing sets are different. For a fair evaluation, we compared our method to many state-of-the-art supervised and unsupervised methods. For the supervised methods, we compared to FMNet [13], GeomFMaps [11], GeomFMaps + DiffusionNet [19] and LIE [75]. For the unsupervised methods, we compared our method to WSupFMNet [39], Smooth Shells [18], Deep Shells [17], NeuroMorph [82] and WSupFMNet + DiffusionNet [19].

As shown in Fig. 4, our method achieves state-of-the-art results among unsupervised methods and supervised methods (we follow the standard Princeton protocol [83] for evaluation). The majority of unsupervised methods and supervised methods fail because of the near-isometry assumption they make. Since our method makes no assumptions about the topology of the shape, it can perform very well in difficult scenarios such as non-isometric animals. It should be noted that these results would not be possible without the second stage of our algorithm, which is enabled by NCP. For example, for

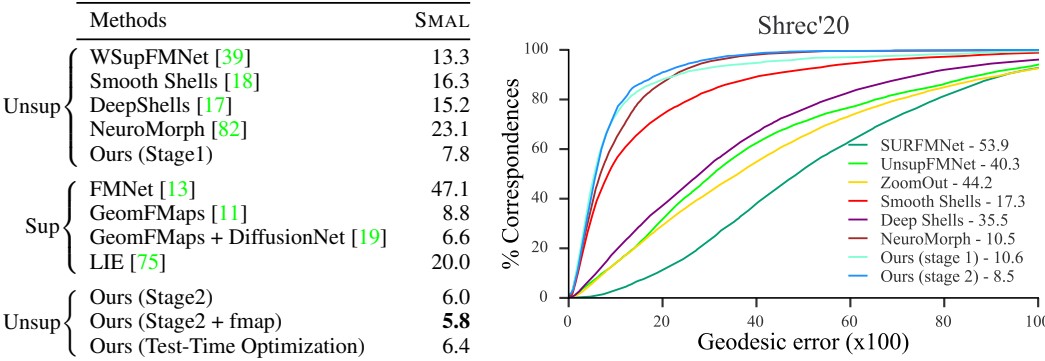

| | Methods | SMAL |
|---|---|---|
| Unsup | WSupFMNet [39] | 13.3 |
| | Smooth Shells [18] | 16.3 |
| | DeepShells [17] | 15.2 |
| | NeuroMorph [82] | 23.1 |
| | Ours (Stage1) | 7.8 |
| Sup | FMNet [13] | 47.1 |
| | GeomFMaps [11] | 8.8 |
| | GeomFMaps + DiffusionNet [19] | 6.6 |
| | LIE [75] | 20.0 |
| Unsup | Ours (Stage2) | 6.0 |
| | Ours (Stage2 + fmap) | **5.8** |
| | Ours (Test-Time Optimization) | 6.4 |

Figure 4: **Non-isometric matching results.** Left: Matching accuracy on the SMAL test set. Values are mean geodesic error $\times 100$ on unit-area shapes. Right: Correspondence quality of different methods on the test set of SHREC'20 Non-Isometric dataset.

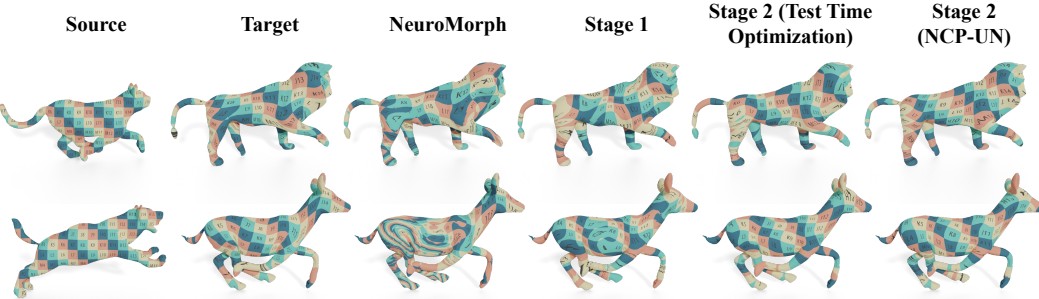

Figure 5: **Qualitative results on SMAL dataset**, showing the maps produced by Stage 1, and refined by Stage 2, either with test time optimization (see Sec. 4.2) or using NCP-UN (see Algorithm 1)

the SMAL dataset, the result of the first stage is **7.8**, and the second stage improves this result to **5.8**, which is a **25%** improvement, achieving state-of-the-art results.

We provide in Fig. 5 some qualitative results showing the performance of our algorithm against the state-of-the-art NeuroMorph method on the SMAL dataset. It can be seen that our method produces visually plausible maps, especially after applying the second stage.

**Test time optimization** To show the utility of test time optimization, as well as the early stopping criteria we introduced, we performed the following experiment. After training the first stage of our algorithm, the trained model was used to predict p2p on the test set of SMAL dataset. Pairs of shapes are then created, and we use our test time optimization method (see Sec. 4.2) to improve upon the maps produced by the first stage. Results are reported in Fig. 4-Left, see row "Ours (Test-Time Optimization)". It can be seen that similar to the previous case, training a random network on top of artifact-laden maps helps to denoise them. We also show in Fig. 5 some qualitative results that illustrate the effect of test time optimization and compare it to NCP-UN.

The results of this experiment confirm our hypothesis that doing the second stage on a collection, instead of individual pairs, provides a second layer of regularization, as the results are better, in addition to improving computational complexity. In fact, the second stage using the whole collection takes around **15 minutes**, meanwhile, doing it on each pair individually takes around **160 minutes**, which represents more than $10\times$ improvement in running time.

## 5.2 Part segmentation transfer

We further validate our approach on the part label transfer task on the PARTNET dataset following the setup of [41]. Results are summarized in Tab. 1, where the average intersection-over-union IOU is reported. Our method outperforms the baselines on 14 out of 16 categories, achieving state-of-the-art results by more than 3% IOU on average. In classes with large intra-class variations such as Lamps, our method outperforms the baselines by up to 8 % average IOU points. In addition to being performant, our method is extremely fast. In fact, training both stages of our method takes on average

Table 1: **Part label transfer** (in the average IOU(%)) for 16 categories in the PARTNET dataset.

| | *pla.* | *bag* | *cap* | *car* | *cha.* | *ear.* | *gui.* | *kni.* | *lam.* | *lap.* | *bik.* | *mug.* | *pis.* | *roc.* | *ska.* | *tab.* | *avg.* |
|---|---|---|---|---|---|---|---|---|---|---|---|---|---|---|---|---|---|
| IDC [43] | 60.1 | 56.2 | 59.7 | - | 72.2 | 45.3 | **81.5** | 66.4 | 42.6 | 88.5 | 40.5 | 87.5 | **66.4** | 37.2 | 50.7 | 70.4 | 61.7 |
| CPAE [41] | 61.3 | 59.3 | 61.6 | - | 72.6 | 55.5 | 78.9 | 71.3 | 53.2 | 89.9 | 55.4 | 86.5 | 66.2 | 40.2 | 61.8 | 72.5 | 65.8 |
| Ours | **63.7** | **66.7** | **68.7** | **57.4** | 80.2 | **59** | 78.8 | **72.5** | **61.9** | **91.4** | **57.2** | **89.5** | 61.4 | **44.2** | **63.6** | **79.2** | **69.2** |

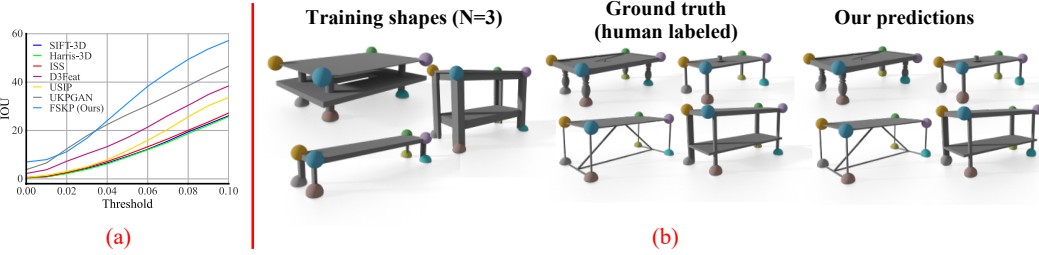

(a)                 (b)

Figure 6: **Few-shot keypoint detection results.**(a) mIoU results of our simple few-shot keypoint detection algorithm on KEYPOINTNET using only 3 labeled examples. Our simple method achieves good results and is on par with or better than recent keypoint detection methods. **(b)** Qualitative results on the table category, showing the shapes used for training (N=3, left), human annotations on the testset (center) as well as our predictions (right) on four examples.

**1 hour** (see the computational specifications in Appendix B ), which is more efficient in comparison to the baselines, where CPAE for example takes around 24 hours to be trained for one category.

## 5.3 Few-shot 3D keypoint detection

We evaluated our few-shot keypoint detection algorithm **FSKD** on the KEYPOINTNET dataset, following the setup of [60], and using only *3 labeled source shapes* (the resulting algorithm is thus a three-shot keypoint detection algorithm). The goal of this task is to predict keypoints that correlate with human annotation. The performance is evaluated by the mean Intersection over Union (mIoU). The intersection is counted if the geodesic distance between a predicted keypoint and its nearest ground truth keypoint is less than some geodesic threshold. The union is simply the union of the detected keypoints and ground truth keypoints.

We compare **FSKD** against multiple baselines, including SIFT-3D [84], HARRIS-3D [85], ISS [86], D3Feat [87], USIP [88] and UKPGAN [60]. Results are summarized in Fig. 6 - left. It can be seen that our simple, correspondence-based method outperforms the baselines tailored specifically for this task. In particular, our method achieves state-of-the-art results for a threshold superior to 0.035, using only **three** labeled shapes. We provide some qualitative results in Fig. 6 - right.

## 6 Conclusion & Limitations

In this work, we showed that using noisy maps as a supervisory neural network training signal for the 3D shape-matching task can lead to significantly higher quality correspondences. We named this effect Neural Correspondence Prior (NCP). Through extensive experiments, we shed light on the properties of NCP and developed a two-stage algorithm for unsupervised 3D shape matching. We demonstrate the effectiveness and generality of our algorithm on a range of shape-matching problems, including unsupervised shape matching on both man-made and organic non-rigid shapes, achieving state-of-the-art results on a wide range of benchmarks.

One limitation of our approach is that it assumes that the p2p maps used to train the second stage of the algorithm still have some structure. In fact, if the unsupervised method used in the first stage fails completely and provides random maps, our method will not converge. This problem can be mitigated by using a good application-specific unsupervised method for the first stage, especially with the recent interest and increase in unsupervised 3D learning methods.

**Acknowledgements** The authors would like to thank Frédéric Chazal and Mathijs Wintraecken for many useful discussions related to Theorem 1. We also acknowledge the anonymous reviewers for their valuable suggestions. Parts of this work were supported by the ERC Starting Grant No. 758800 (EXPROTEA) and the ANR AI Chair AIGRETTE.

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
