# Supplementary Materials for:
# NCP: Neural Correspondence Prior for Effective Unsupervised Shape Matching

**Souhaib Attaiki**
LIX, École Polytechnique, IP Paris
attaiki@lix.polytechnique.fr

**Maks Ovsjanikov**
LIX, École Polytechnique, IP Paris
maks@lix.polytechnique.fr

In this document, we collect all the results and discussions, which, due to the page limit, could not find space in the main manuscript.

Specifically, we first give a background on functional maps in Appendix A. Next, the implementation details are provided in Appendix B. The implementation details of our few-shot keypoint detection algorithm are presented in Appendix C. In Appendix D, we provide a complete formulation and proof of the theorem we introduced in Sec. 3.2 of the main text. In Appendix E, we provide an experiment to verify the conditions of the aforementioned theorem. We present a more in-depth analysis of the Neural Correspondence Prior (NCP) effect in Appendix F. Additional quantitative and qualitative results for the shape matching on man-made data tasks are included in Appendix G. An experiment showing the performance of our method in the case of near isometric data is presented in Appendix H. Finally, we discuss the societal impact of our work in Appendix I.

## A    Background on functional maps & Notation

Our work uses the functional map framework as a first estimator for p2p maps, and multiple losses on point-to-point (p2p) maps to learn robust features that allow extracting good correspondences using the nearest neighbor in feature space. We provide a brief overview in the next section.

**Functional maps**    The functional map (fmap) framework was used for the first stage of our **NCP-UN** algorithm. For this, we follow the general strategy of recent fmap-based techniques [1, 2, 3, 4, 5], as follows: given source and target shapes $\mathcal{M}$ and $\mathcal{N}$, represented as either triangular meshes or point clouds, having $m$ and $n$ vertices respectively, we pre-compute their Laplace-Beltrami operator [6], and store their first $k$ eigenfunctions in the matrices $\Phi_{\mathcal{M}} \in \mathbb{R}^{m \times k}$ and $\Phi_{\mathcal{N}} \in \mathbb{R}^{n \times k}$ respectively. Using a siamese network $\mathcal{F}_{\theta}$, we compute for each shape a $d$-dimensional descriptor $\mathcal{F}_{\theta}(\mathcal{M}) = \mathbf{F} \in \mathbb{R}^{m \times d}$ and $\mathcal{F}_{\theta}(\mathcal{N}) = \mathbf{G} \in \mathbb{R}^{n \times d}$ respectively. These descriptors are then projected to the spectral domain to form the spectral features $\mathbf{A} = \Phi_{\mathcal{M}}^{\dagger} \mathbf{F}$ and $\mathbf{B} = \Phi_{\mathcal{N}}^{\dagger} \mathbf{G}$ ($\bullet^{\dagger}$ is the Moore pseudo-inverse). A functional map is then computed by solving the following linear system:

$$C_{opt} = \arg\min_{C} \|C\mathbf{A} - \mathbf{B}\| + \lambda \|C\Delta_{\mathcal{M}} - \Delta_{\mathcal{N}} C\|. \tag{1}$$

where $\Delta_{\mathcal{M}}, \Delta_{\mathcal{N}}$ are diagonal matrices of Laplace-Beltrami eigenvalues of the corresponding shapes and $\lambda$ is a scalar hyper-parameter.

Following the unsupervised literature [4, 3, 5], the siamese network $\mathcal{F}_{\theta}$ is trained by imposing structural properties on the fmap $C$ such as bijectivity and orthogonality on the shape pairs in the training set. In fact, given the fmap $C_{\mathcal{MN}}$ from $\mathcal{M}$ to $\mathcal{N}$, and $C_{\mathcal{NM}}$ from $\mathcal{N}$ to $\mathcal{M}$, the bijectivity loss is formulated as $\|C_{\mathcal{MN}} C_{\mathcal{NM}} - \mathbb{I}_k\|_2^2$, and the orthogonality loss is $\|C_{\mathcal{MN}}^{\top} C_{\mathcal{MN}} - \mathbb{I}_k\|_2^2$. $\mathbb{I}_k$ denotes the identity matrix of size $k$ and $\bullet^{\top}$ is the matrix transpose operator.

36th Conference on Neural Information Processing Systems (NeurIPS 2022).

**Feature learning** The goal of feature learning is to learn robust descriptors that can allow direct nearest-neighbor matching in the descriptor space. In this work, we use two losses: PointInfoNCE and the LIE loss.

PointInfoNCE [7] is a contrastive loss such that, given a set of matched points $\mathcal{P}$, and two features of dimension $s$, it is formulated as follows:

$$\mathcal{L}_{\text{NCE}} = - \sum_{(i,j) \in \mathcal{P}} \log \frac{\exp(d(\mathbf{F}_i, \mathbf{G}_j)/\tau)}{\sum_{(\cdot,k) \in \mathcal{P}} \exp(d(\mathbf{F}_i, \mathbf{G}_k)/\tau)} \tag{2}$$

$$d(\mathbf{F}_i, \mathbf{G}_j) = \|\mathbf{F}_i - \mathbf{G}_j\|_2^2 \tag{3}$$

where $\tau$ is a temperature parameter, and $d(\cdot, \cdot)$ is the Euclidean distance between the two features. In all our experiments, we took $\tau = 0.07$. The purpose of this loss is to force the distance between the features of the matched points to be minimized, while this distance must be maximized between the unmatched points. The NCE loss is applied to each point individually and thus cannot penalize the overall consistency of the matches.

To remedy this, especially when the number of vertices of the shapes is moderate, as is the case for sparse point clouds, we use the LIE loss introduced in [8]. Given the extracted features $\mathbf{F}$ and $\mathbf{G}$, and the coordinate *xyz* of the shape $\mathcal{N}$ represented by the matrix $\mathbf{N} \in \mathbb{R}^{n \times 3}$, we first compute the soft correspondences matrix $S_{\mathcal{MN}}$, and then formulate the LIE loss as follows:

$$\mathcal{L}_{\text{LIE}} = \|S_{\mathcal{MN}}\mathbf{N} - \Pi_{\mathcal{MN}}^{gt}\mathbf{N}\|_2^2, \tag{4}$$

$$(S_{\mathcal{MN}})_{ij} = \frac{\exp(-\|\mathbf{F}_i - \mathbf{G}_j\|_2)}{\sum_{k=1}^{n} \exp(-\|\mathbf{F}_i - \mathbf{G}_k\|_2} \tag{5}$$

where $\Pi_{\mathcal{MN}}^{gt}$ is the ground truth p2p correspondence matrix. This loss forces the soft correspondences matrix to be as close as possible to the ground truth map, by forcing their action (pull-back) on the shape coordinates, thus taking into account the geometry of the shape. Indeed, erroneous predictions that are geometrically close to the ground truth are penalized less than those that are far from it in terms of L2 distance.

## B   Implementation details

In Sec. 3.1 in the main text, we used a randomly initialized DiffusionNet [5] network to predict features on the test set of FAUST-Remeshed (FAUST) and SCAPE-Remeshed (SCAPE) datasets [9]. For that, we used the publicly available implementation of DiffusionNet released by the authors [1]. Unless specified otherwise, all our experiments on 3d-triangular meshes use 4 DiffusionNet blocks of width 128, the input to the network is the XYZ coordinates of the shape. For the competing features, both the Heat Kernel Signature (HKS) [10] and the Wave Kernel Signature (WKS) [11] were sampled at 100 values of energy t, logarithmically spaced in the range proposed in their respective original papers. SHOT descriptors [12] are 352-dimensional, and we used the implementation provided by the PCL library [13]. For the Laplace-Beltrami computation, we used the discretization introduced in [6] for both 3D meshes and point clouds.

In Sec. 3.2 of the main text, we train a DiffusionNet to produce feature embeddings that will induce the maps used for supervision, using the NCE loss. For this experiment and all the following learning experiments, ADAM optimizer [14] was used with a learning rate of 0.001.

In Sec. 5.1.1 and 5.2 of the main text, we applied our **NCP-UN** algorithm on the KEYPOINTNET [15] and the PARTNET [16] datasets. For the first stage, we used the unsupervised geometric functional map from [5] but used PointMLP [17] as a feature extractor, instead of DiffusionNet, as it is better suited to the point cloud context. We used the default segmentation configuration provided by the authors [2]. In Eq. (1), we take $\lambda = 0$. The network is trained using the bijectivity loss presented in Appendix A, as well as a new unsupervised loss based on the chamfer distance that we introduced.

---

[1] https://github.com/nmwsharp/diffusion-net
[2] https://github.com/ma-xu/pointMLP-pytorch

Indeed, we compute the chamfer distance between the source shape's XYZ coordinates, and a new version of this shape created by transferring its coordinates into the spectral space of the target shapes and back again. Formally, using the same notation as above, it is as follows:

$$\mathcal{M}_{source} = \Phi_{\mathcal{M}}\Phi_{\mathcal{M}}^{\dagger}\mathbf{M} \tag{6}$$

$$\mathcal{M}_{target} = \Phi_{\mathcal{M}}C_{\mathcal{N}\mathcal{M}}C_{\mathcal{M}\mathcal{N}}\Phi_{\mathcal{M}}^{\dagger}\mathbf{M} \tag{7}$$

$$L_{\text{chamfer}}(\mathcal{M}_{\text{source}}, \mathcal{M}_{\text{target}}) = \sum_{m\in\mathcal{M}_{\text{source}}}\min_{n\in\mathcal{M}_{\text{target}}}\|m-n\|_2 + \sum_{n\in\mathcal{M}_{\text{target}}}\min_{m\in\mathcal{M}_{\text{source}}}\|m-n\|_2 \tag{8}$$

For the second stage, we used the LIE loss, and we trained a randomly initialized PointMLP, which has the same architecture as the first stage, to produce feature embeddings that induce the input maps.

In Sec. 5.1.2 of the main text, **NCP-UN** was applied on the SMAL [18, 19] and SHREC'20 [20] datasets. The first stage was performed using the unsupervised geometric functional map [5] with the DiffusionNet backbone while applying the bijectivity and orthogonality losses, as described above in Appendix A. In Eq. (1), we set $\lambda = 10^{-3}$ for SMAL, and $\lambda = 0$ for SHREC'20. The second stage was performed using the LIE loss and the same backbone as the first stage. P2P maps were extracted using either the nearest neighbor in the space of features or using the functional map pipeline. In fact, given two feature embedding for two shapes $\mathcal{M}$ and $\mathcal{N}$, we compute the functional map $C_{\mathcal{M}\mathcal{N}}$ from $\mathcal{M}$ to $\mathcal{N}$ using Eq. (1), and then convert it to a p2p map $T_{fmap} : \mathcal{N} \to \mathcal{M}$ using (borrowing the same notation from Appendix A):

$$T_{fmap}(y) = \arg\min_{x}\|(\Phi_{\mathcal{N}})_y - (\Phi_{\mathcal{M}}C_{\mathcal{M}\mathcal{N}}^{\top})_x\| \tag{9}$$

For the test time optimization experiment in Sec. 5.1.2 of the main text, after extracting the maps predicted by the first stage, we construct pairs between all the shapes of the test set, and for each pair, we train a randomly-initialized DiffusionNet network using the NCE loss and Adam optimizer, to produce feature embedding that induces the map from stage 1. We stop the training when the cyclic loss (see Sec. 4.2 of the main text) stops improving giving a patience period of 100 optimization iterations.

In all the experiments of Sec. 5 of the main text, except for test time optimization, data augmentation was used. In particular, we augment the training data on the fly by randomly rotating the input shapes, applying random scaling in the range [0.9, 1.1], and jittering the position of each point by Gaussian noise with zero mean and 0.01 standard deviation.

**Computational specifications** All our experiments are executed using Pytorch [21], on a 64-bit machine, equipped with an Intel(R) Xeon(R) CPU E5-2630 v4 @ 2.20GHz and a RTX 2080 Ti Graphics Card. For all competing methods, we use the original code released by the authors and apply the best parameters reported in the respective papers. As mentioned in the main paper, we will release our complete implementation to ensure the full reproducibility of all of our results.

## C   Implementation details on FSKD

Below we provide the implementation details on our Few-Shot Keypoint Detection method (FSKD), described in Sec. 5.3 of the main manuscript. As mentioned in that section, **FSKD** is composed of three steps: **1.** Detection of potential keypoints by transferring keypoints from labeled shapes, **2.** Filtering to remove keypoints that are likely not to exist on the target shape, **3.** Combination: merge transferred keypoints if multiple points on the target shape are assigned to the same keypoint ID.

**Step 1.** is done using our established maps predicted by **NCP-UN**.

For **step 2.**, we compute the cycle consistency loss of transferred keypoints and only keep the ones that are below a predetermined threshold. I.e., given a pair of shapes $\mathcal{M}, \mathcal{N}$, and two maps between them $\Pi_{\mathcal{M}\mathcal{N}}$ and $\Pi_{\mathcal{N}\mathcal{M}}$ computed via nearest neighbor matching between their feature embeddings, and represented as binary matrices, the cycle consistency loss of keypoints $i$ is computed as $l_i = \|(X_{\mathcal{N}})_i - (\Pi_{\mathcal{N}\mathcal{M}}\Pi_{\mathcal{M}\mathcal{N}}X_{\mathcal{N}})_i\|_F^2$, where $X_{\mathcal{N}}$ is the matrix of XYZ coordinates of the source shape $\mathcal{N}$. If $l_i$ is bigger than a predefined threshold $\nu$, the keypoint $i$ is considered not to exist on the target shape. In our experiments, we take $\nu = 0.05$.

Concerning **step 3.**, we perform a spatially weighted average of the different filtered keypoints if there are many. We associate for each keypoint $i$ the weight $w_i = \frac{D_i}{\sum_j D_j}$, where $D_i = \exp(-\frac{l_i}{\sigma})$. We use $\sigma = 0.01$.

## D   Proof of smoothness of maps produced by smooth networks

In Sec. 3.2 of the main text, we briefly mentioned a theorem that states that maps based on the nearest neighbor between smooth features are smooth. In this section, we formally restate it and provide a proof.

**Theorem 1.** *Let $\mathcal{M}$ and $\mathcal{N}$ be two compact smooth surfaces (smooth manifolds of dimension 2). Let $\mathbf{M}$ and $\mathbf{N}$ be their embeddings in $\mathbb{R}^d$, given by some functions: $\psi : \mathcal{M} \to \mathbb{R}^d$ and $\phi : \mathcal{N} \to \mathbb{R}^d$, so that $\mathbf{M} = \psi(\mathcal{M})$ and $\mathbf{N} = \phi(\mathcal{N})$. Suppose that $\psi$ and $\phi$ are both smooth and injective. Then up to arbitrarily small perturbations of $\phi, \psi$, the map $T_{nn} : \mathcal{M} \to \mathcal{N}$ given by $T_{nn}(x) = \arg\min_{y \in \mathcal{N}} \|\psi(x) - \phi(y)\|$ must be smooth up to sets of measure 0 on $\mathcal{M}$.*

*Proof.* First, note that the distance function to an embedded manifold is smooth almost everywhere. Indeed, it is well-known that if $\mathbf{N}$ is a $C^k$-continuous manifold embedded in $\mathbb{R}^d$, then the distance function to $\mathbf{N}$ must be at least $C^k$ continuous on the complement of the medial axis of $\mathbf{N}$. I.e., let $d_{\mathbf{N}} : \mathbb{R}^d \to \mathbb{R}^d$ be given by $d_{\mathbf{N}}(x) = \arg\min_{y \in \mathbf{N}} \|x - y\|$. Let $\mathrm{cut}(\mathbf{N})$ denote the medial axis of $\mathbf{N}$, which is defined as the set of points in $\mathbb{R}^d$ with more than one nearest neighbor to $\mathbf{N}$. I.e., $\mathrm{cut}(\mathbf{N}) = \{x \in \mathbb{R}^d \mid \exists\, y_1, y_2 \in \mathbf{N}, y_1 \neq y_2, \text{s.t.} \|x - y_1\| = \|x - y_2\| = \min_{y \in \mathbf{N}} \|x - y\|\}$. Then, $d_{\mathbf{N}}$ is at least $C^k$ continuous on $\mathbb{R}^d \backslash \mathrm{cut}(\mathbf{N})$ (see Lemma 2.5 in [22], and [23, 24] for this and related results).

It remains to prove that up to arbitrarily small perturbations of $\phi$ and $\psi$ the intersection between $\mathbf{M}$ and $\mathrm{cut}(\mathbf{N})$ has measure zero on $\mathbf{M}$. For this, we first use the fact that the medial axis of compact subanalytic manifolds is also subanalytic [25]. This means that the medial axis can be stratified (decomposed into a finite union of submanifolds of dimension $d - 1$ in $\mathbb{R}^d$). Since (by Stone-Weierstrass's theorem) subanalytic manifolds are dense in the space of smooth manifolds, up to an arbitrarily small perturbation of $\phi$, $\mathbf{N}$ is subanalytic. Finally, the intersection between $\mathbf{M}$ (which is an embedded manifold of dimension 2) and any manifold of dimension $d - 1$ by Thom's transversality theorem [26, 27], must generically be of measure zero on $\mathbf{M}$ and thus on $\mathcal{M}$.                               $\square$

## E   Verification of assumptions of Theorem 1

The main motivation behind Theorem 1 is to highlight the fact that, given smooth feature embeddings, if we add the injectivity condition, the maps extracted using the nearest neighbors in the feature space tend to be smooth. As smoothness is a generally desirable property, and the frequency bias shown in prior works, such as [28] suggests that neural networks are biased towards low frequency (and thus smooth) functions, we provide this result as a partial explanation for the Neural Correspondence Prior, which we have observed, for the first time, in our work.

Regarding injectivity, although such a property is not trivial, and might need to be specifically enforced (using, for example, invertible networks [29]). However, we observe that when training a network using contrastive learning, such as NCE loss, the latter forces the networks to produce embeddings that are unique for each point, in order to minimize the NCE loss, which can be considered a form of infectivity.

We include in this section the results of an experiment we performed in order to examine the smoothness and injectivity of a randomly initialized network. We start by computing the feature embeddings produced by a randomly initialized DiffusionNet network for the 20 test shapes in the FAUST [9] dataset.

To evaluate the smoothness of the embedding, we compute the standard Dirichlet energy of both the embedding produced by the network and the original embedding of the shape in $\mathbb{R}^3$ (the 3D coordinates of the shape's vertices), using the following formula:

$$E_{Dirichlet}(G) = \frac{1}{n} \sum_{i=1}^{n} \frac{G_i^\top W G_i}{G_i^\top A G_i} \tag{10}$$

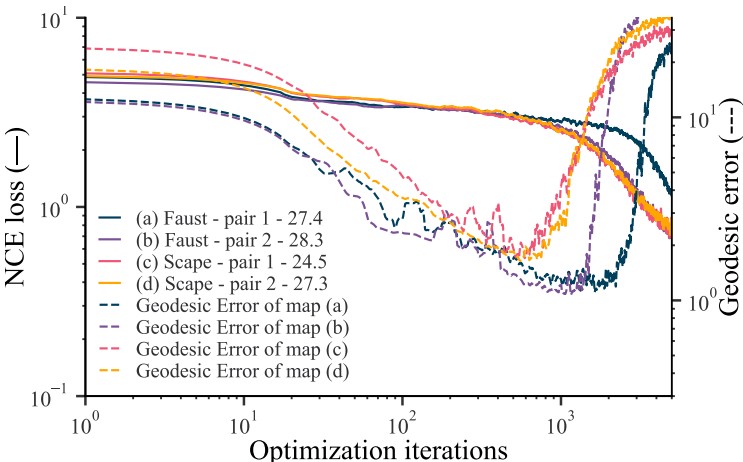

Figure 1: Learning curves showing the NCE loss and the geodesic error for different shape pairs from FAUST and SCAPE datasets. Numbers in parentheses represent the geodesic error of the input maps. Observe how the effect of NCP doesn't depend on the shape pair.

where $G$ is the considered embedding of dimension $n$, while $W$ and $A$ are, respectively, the standard stiffness and mass matrices, computed using the classical cotangent discretization scheme of the Laplace-Beltrami operator [30]. We compute the Dirichlet energy over the whole test set and find it equal to **47.2** (the average) for coordinate functions (original shape embeddings in $\mathbb{R}^3$), while it is equal to **13.1** for embeddings produced by the network. This shows that the latter are smoother than embeddings in the original space.

For injectivity, we compute for each point of the embedding, the distance to its nearest neighbor, and took the minimum across all points of a shape. To make the comparison between the original embedding and the embedding produced by the network fair, we normalize both embeddings to the unit sphere. We find that on average, the minimum distance between points in the original 3D space is **0.0004**, while for the feature embeddings given by the network, it is equal to **0.0015**. This shows that the network is injective and that distances between points are larger than in the original domain.

## F    Neural Correspondence Prior

In Sec. 3 of the main text, we demonstrated the effect of Neural Correspondence Prior (NCP), on a pair of shapes, using the DiffusionNet network. The objective of this section is to show that the NCP is independent of the choice of the shape pair, the choice of the p2p loss, the choice of the architecture, and finally the choice of the first stage.

### F.1    Independence from shape pairs

In order to show that the NCP effect demonstrated in Figure 1 of the main text does not depend on the chosen shape, we redid the same experiment using new random pairs from FAUST and SCAPE datasets [9]. Following the same setup of Sec. 3.2 of the main text, given a pair of shapes, we corrupt the ground truth map between them with 50% noise, and then train a randomly initialized DiffusionNet to produce feature embeddings that overfit the noisy map. Examples of four different pairs are shown in Fig. 1. We see that the same effect is always present, i.e., the network resists overfitting the noisy maps (high noise impedance), and the intermediate maps during optimization are of high quality, compared to the noisy maps used as supervision.

### F.2    Independence from the loss function and network architecture

In order to examine the independence of NCP on both the network architecture and on the p2p loss used in stage 2 of **NCP-UN**, the following experiment was performed on the Chair subset of the KEYPOINTNET [15] dataset. Since the latter doesn't provide dense ground truth p2p maps between the shapes, we took the p2p maps produced by the first stage of our algorithm (see Sec. 5.1.1 of the

Table 1: **Ablation study on the loss function and network architecture**. We show the results of **NCP-UN** on the Chair subset of the KEYPOINTNET dataset [15] with different losses and network architectures. It appears that multiple network architectures and losses improve the maps produced by the first stage, proving that NCP is not tied to a single method choice.

| Metrics | Input maps | PointMLP | | | DGCNN | LIE | ResidualMLP |
| | | LIE | NCE | FMAP | | PointNet++ | |
|---|---|---|---|---|---|---|---|
| Geodesic error ($\times 100$) | 9.5 | 5.1 | 5.1 | 5.2 | 5.0 | 5.3 | 8.1 |
| Map smoothness | 234.9 | 10.3 | 10.1 | 9.9 | 11.3 | 9.7 | 8.3 |

Table 2: **Ablation study on the method used for Stage 1**. We show the results of **NCP-UN** on the SMAL dataset using multiple methods for Stage 1. Values are mean geodesic error ×100 on unit-area shapes. It can be seen that the NCP effect still applies despite the chosen method for Stage 1.

| Stage / Method | NeuroMorph [36] | Smooth Shells [34] | Deep Shells [35] | Unsup GeomFMaps [5] |
|---|---|---|---|---|
| Stage 1 | 23.1 | 16.3 | 15.2 | 7.8 |
| Stage 2 | 7.2 (+68%) | 8.1 (+50%) | 7.3 (+52%) | 5.8 (+25%) |

main text), and perturbed them with noise. These noisy maps are used as inputs to the second stage of the **NCP-UN** algorithm. Differently, from the experiment performed in Sec. 5.1.1 of the main text, here, we choose different network architectures and losses for training the second stage. In particular, we consider the NCE loss, the LIE loss, and the supervised FMAP loss from [1]. Concerning the network architectures, we considered the ResidualMLP network introduced in [31], PointNet++ [32], and DGCNN [33]. For the network architectures, we used the official implementation provided by the authors.

In addition to measuring the geodesic error produced by the maps predicted by the second stage, we also measure their smoothness. Given two shapes $\mathcal{M}, \mathcal{N}$, and a map between them $T_{\mathcal{M}\mathcal{N}}$ represented as a binary matrix $\Pi_{\mathcal{M}\mathcal{N}}$, we compute the smoothness of the latter based on the Dirichlet energy using the following (see [30] for more details):

$$E_{smoothness}(T_{\mathcal{M}\mathcal{N}}) = \sum_{(u,v)\in\mathcal{E}_{\mathcal{M}}} w_{uv}\|\psi_{\mathcal{M}\mathcal{N}}(u) - \psi_{\mathcal{M}\mathcal{N}}(v)\|_2^2 \tag{11}$$

$$\psi_{\mathcal{M}\mathcal{N}} = \Pi_{\mathcal{M}\mathcal{N}}X_{\mathcal{N}} \tag{12}$$

where $X_{\mathcal{N}}$ is the matrix of XYZ coordinates of shape $\mathcal{N}$, $\mathcal{E}_{\mathcal{M}}$ is the set of all the edges of the triangular mesh, and $w_{uv}$ are the stiffness weights of the cotangent Laplacian [30] for shape $\mathcal{M}$.

Results of this experiment are summarised in Tab. 1. It can be seen that all the losses and network architectures perform well and manage to improve the noisy input maps. One can also notice the smoothing effect of NCP. In fact, the second stage of the network produces feature embeddings that result in p2p maps that are not only geometrically correct but also remove noise and outliers from the input maps. It is worth noting that although the ResidualMLP architecture is weak, since it is applied on each vertex individually, and has no global shape awareness, we note that even if the geodesic error is bad, the network tends to produce smooth embeddings.

### F.3 Independence from the first stage

As we stated in the main paper, our **NCP-UN** algorithm is independent of the first stage, i.e the method used to extract the artifact-laden p2p is not crucial, and the NCP effect still applies. To demonstrate this, we followed the same setup of Sec. 5.1.2 of the main text, and we perform the same experiment on the SMAL dataset but using different methods for stage 1. In particular, we consider as stage 1 the Unsupervised geometric functional map (Unsup GeomFMaps) [5], Smooth Shells [34], Deep Shells [35] and NeuroMorph [36]. Results are summarized in Tab. 2. As can be seen, despite the method used in the first step, the NCP still applies, and a significant improvement in results is observed, up to 68% improvement.

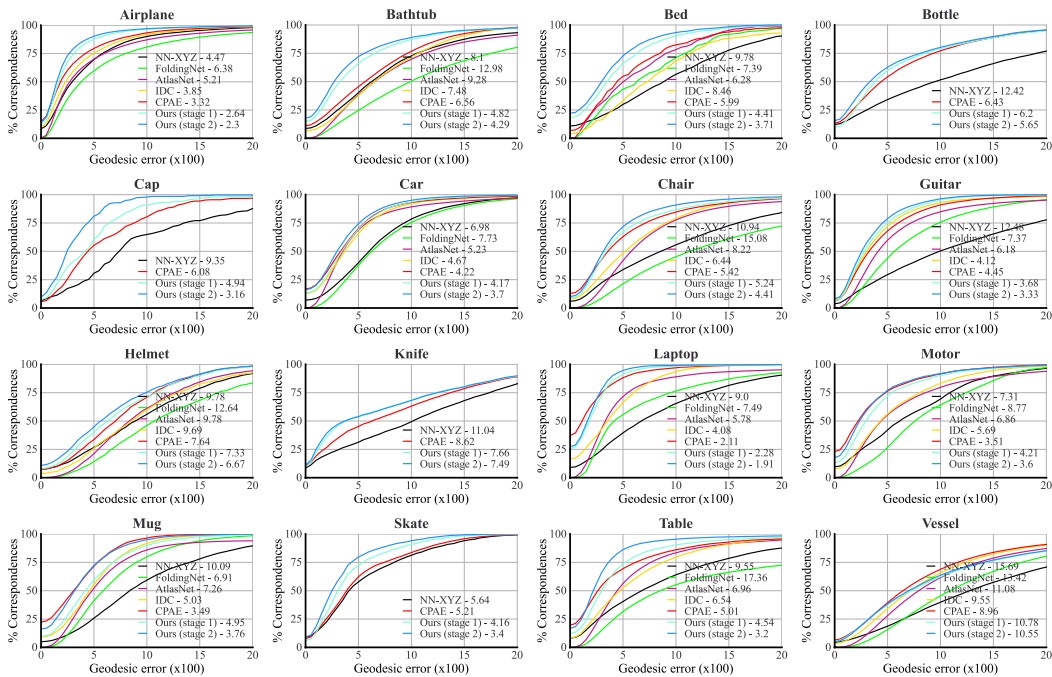

Figure 2: **Correspondence accuracy on the KEYPOINTNET dataset**. It can be seen that the second stage of our algorithm always improves upon the first stage due to the NCP effect.

## G   Shape matching on man-made data

In Sec. 5.1.1 of the main text, we applied our **NCP-UN** algorithm on the KEYPOINTNET dataset, and only provided quantitative results for 4 classes due to the page limit. We provide in Fig. 2 quantitative results for all the 16 classes, in addition to the results of the first stage of our algorithm, i.e the unsupervised method described above in Appendix B. For the bottle, cap, knife, and skate categories, we only compared them to the best-performing baseline. It can be seen that our method achieves state-of-the-art results in 13 out of 16 classes on this benchmark, with an impressive improvement in some classes such as cap and table. We also see that the second stage of **NCP-UN** always improves the result provided by the first stage, which again demonstrates the role of the NCP effect, without which the SOTA result would not be achieved, see for example the laptop category.

Additionally, we provide in Fig. 3 some qualitative results, showing the p2p maps produced by both stages of **NCP-UN**, visualized using texture transfer, as well as the usage of these maps to transfer keypoints between shapes. It can be seen that the keypoint transferred by the maps of stage 2 are more accurate.

## H   Shape matching on non-rigid near-isometric data

In our work, we focus on difficult non-rigid non-isometric datasets, where existing methods tend to fail. This is because on near-isometric datasets such as FAUST or SCAPE [9], current unsupervised methods can exploit the assumption of near-isometry and achieve good results.

Nevertheless, we include a comparison of our method on the FAUST and SCAPE datasets, using the same train/test split used in all previous works (e.g. [1]). The notation X on Y means the method is trained on X and tested on Y.

As input for our Stage 2, we use WSupFMNet + DiffusionNet [5] (referred to as Stage 1 in the table), which is the same method used in the main manuscript. For our method (Stage 2), we use the same implementation as the one used for the SMAL dataset, described in Sec. 5.1.2 of the main manuscript.

As shown in Tab. 3, even on a near-isometric dataset, our Stage 2 improves upon the initial maps in all categories, by **18.4%** on average. Nevertheless, we remark that in such settings it is more

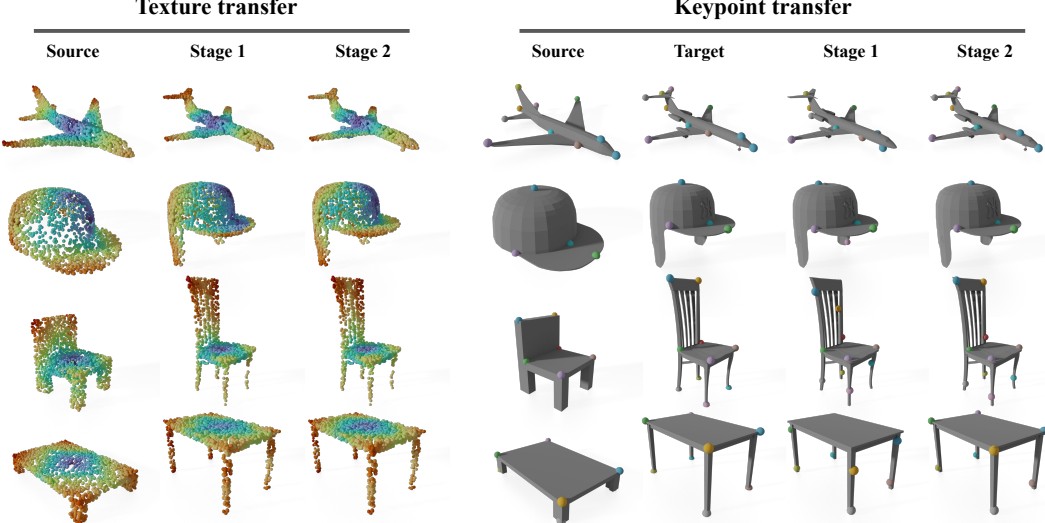

| Texture transfer | | | Keypoint transfer | | | |
| Source | Stage 1 | Stage 2 | Source | Target | Stage 1 | Stage 2 |

Figure 3: **Qualitative results on the KEYPOINTNET dataset** using four categories: airplane, cap, chair, and table. Both the point-to-point map as well as the keypoint transfer are presented. Each row contains the p2p maps, ground truth keypoint annotations, and predictions made by the first and the second stage of our algorithm.

Table 3: **Results of NCP-UN on near-isometric data**, using the FAUST (**F**) and SCAPE (**S**) datasets. Values are mean geodesic error ×100 on unit-area shapes. It can be seen that the NCP effect still applies even in the case of near-isometric shapes.

| Methods | *F on F* | *S on S* | *F on S* | *S on F* |
| --- | --- | --- | --- | --- |
| Stage 1 | 3.8 | 4.4 | 4.8 | 3.6 |
| Stage 1 + ZoomOut | 1.9 | 2.6 | 2.7 | 1.9 |
| NCP (Ours - Stage 2) | **3.0** | **3.5** | **4.2** | **2.9** |
| NCP (Ours - Stage 2) + ZoomOut | 1.9 | 2.4 | 2.6 | 1.9 |

advantageous to use specialized methods, such as ZoomOut [37], that directly exploit the near-isometry assumption.

# I   Societal impact

Efficient methods for shape matching and analysis have an immediate impact in many areas of science and engineering from medical imaging (for instance detecting anomalies, and performing follow-up analysis) to shape recognition and classification in areas such as computational biology, archaeology, and paleontology to name a few. Our approach can immediately be adapted to such diverse scenarios, due to its strong generalization power, and its generic unsupervised nature, especially in domains where acquiring data is easy, but labeling it is very expensive, e.g in structural or molecular biology. Our work also opens up important avenues for future research, as it can enable geometric deep learning methods without the need to label large-scale datasets, thus potentially allowing small labs to conduct research in this area without the need to hire many annotators, which is an expensive task. Finally, our method paves the way for more accurate results, helping to improve our understanding in many fields, such as biology where shape-matching techniques are used to analyze gene expression patterns to understand the cause of many human syndromes [38]. Since our method attempts to solve a fundamental problem in computer graphics and computer vision, we do not expect negative results. However, one should note that highly accurate shape correspondence methods might have possibly problematic uses, e.g., in surveillance applications, although we advocate against such uses of our technique.