# OpenReview forum: "NCP: Neural Correspondence Prior for Effective Unsupervised Shape Matching"
_NeurIPS.cc/2022/Conference — NeurIPS 2022 Accept_

### Official Review · Reviewer_2gYm · 2022-06-24

**Rating:** 3
**Confidence:** 4
**Soundness:** 3 good
**Presentation:** 3 good
**Contribution:** 2 fair

**Summary:**

An unsupervised shape correspondence method is proposed :

1. Inputs : training set of shapes (without ground truth)
2. Train an off-the-shelf unsupervised matching "teacher" network on the training set
3. Train a "student" network from scratch on the training set in a supervised manner using the "teacher" predictions as ground truth
4. Use the "student" network to establish correspondences between unseen shape pairs

This method is motived by two empirical observations :

1. "Neural bias" : It experimentally shows that features extracted with a randomly initialized DiffusionNet produce good matching performances on FAUST remeshed and SCAPE-remeshed datasets .
2. "Neural Correspondence Prior" : It experimentally shows that the "Deep Image Prior" holds for a shape correspondence problem.

The proposed method either outperforms or is on par with the sota methods on the KeyPointNet dataset, the SHREC20 dataset,  the SMAL dataset and the PartNet dataset.

**Questions:**

Please see "Weaknesses" above.

**Limitations:**

-

**Strengths And Weaknesses:**

### Strengths

1. Extending the "Deep Image Prior" to shape matching is an interesting idea.
2. The paper is well written.
3. The overall approach is simple.

### Weaknesses

#### 3.1 Neural bias for pointwise features

1. This part of the paper experimentally shows that features extracted with a randomly initialized diffusionNet produce good matching performances on FAUST remeshed and SCAPE-remeshed datasets. I believe the results of a randomly initialized pointwise MLP should be added, otherwise the section title should be changed to "diffusionNet bias for pointwise features".
2. The link between this neural bias (i.e the fact that an untrained network produces good matches) and the proposed method is unclear to me, since both the teacher and the student networks are trained.

#### 4.1 NCP within a shape collection

1. Why not using the standard terminology teacher network / student network ? I think it would make the whole paper easier to read.
2. I was expecting the teacher network to be an untrained diffusionNet, especially after reading section 3.1. It seems to me that section 3.1 is not tightly connected to the rest of the paper.

#### 5.1.1 Correspondences on Point Clouds

1. The proposed method outperforms the sota methods but I could not find the performances of the teacher network (i.e unsupervised functional map method + PointMLP backbone).

#### 5.1.2 Correspondences between non-rigid meshes

1. Table 2 shows that the proposed method only slightly outperforms the DiffusionNet-based methods, for instance "WSupFMNeT + DiffusionNet" is an unsupervised method that is only slighly outperformed by the proposed method.
2. Figure 4 shows the results of NeuroMorph which is the worst unsupervised method in Table 2. Why not showing the results of "WSupFMNeT + DiffusionNet" instead ?

Overall the proposed method is probably interesting for the shape matching community but the novelty is limited. Moreover, it seems the proposed method only slightly outperforms Diffusion-based methods.

---

> ### Author Response · Authors · 2022-08-02
> **Responses to Reviewer 2gYm (Part 1/2)**
>
> We thank the reviewer for their detailed and thoughtful comments and suggestions. We invite the reviewer to consider the general feedback that we provided to all reviewers in the "High Level Summary" answer. Below are our answers to the reviewers’ questions. Due to the 5000 character limit, and in order to fully address the reviewers’ comments, we have divided our response into two parts.
>
> **Q1. Neural bias with pointwise MLP**
>
>  Thank you for the suggestion. Please
> note that the main goal of Section 3.1 is to highlight our observation
> that the structure of the network, even when untrained, provides a prior
> for the produced features. We used DiffusionNet to illustrate this
> point, following the Deep Image Prior work \[1\], which only
> demonstrated this effect on one, commonly used, state-of-the-art
> architecture.
>
> At the same time, we stress that we only used this observation as a
> *motivation* for our main proposal (Section 4), built from the *Neural
> Correspondence Prior,* described in Section 3.2.
>
> We have evaluated the Neural Correspondence Prior and our proposed
> algorithm with five different backbone architectures, including
> DiffusionNet, PointMLP, DGCNN, PointNet++, and ResidualMLP, which is a
> pointwise MLP. Please see Table 1 of the Supp. Materials.
>
> We will be happy to reword and possibly rename Section 3.1, as
> requested.
>
> \[1\] Dmitry Ulyanov, Andrea Vedaldi, and Victor Lempitsky. Deep image
> prior. arXiv:1711.10925, 2017.
>
> **Q2. Link between Neural bias with pointwise MLP**
>
> As mentioned above, we
> view the neural bias as a general motivation to our approach,
> highlighting the fact that the structure of neural networks imposes a
> particular prior on the types of features that they produce. We used the
> neural prior to illustrate this effect as it is especially striking when
> the networks are untrained. However, we do not use it in our
> experimental results, which are based on Algorithm 1, described in
> Section 4. We will clarify this in the final version.
>
> **Q3. Terminology of teacher / student networks**
>
> We avoided this
> terminology, as it is borrowed from the knowledge distillation
> community, where the goal is to transfer the “knowledge” of a bigger
> network (in term of size and parameters) (teacher) to a smaller network
> (student).
>
> This is not the case for our method, and using such a terminology can
> potentially be misleading, especially to readers from other communities.
> First, Stage 1 of our algorithm uses an off-the-shelf unsupervised shape
> matching method that can be either learning based (such as unsupervised
> GeomFMaps) or even axiomatic (such as Smooth Shells, see Table 2 of the
> Supp. Materials), and no condition on the size or type of a network is
> made.
>
> Second, the purpose of the Stage 1 is to produce a set of approximate
> maps, which are then refined in Stage 2. To the best of our knowledge, while
> in the student/teacher paradigm, the student is trained to learn exactly
> the same predictions by the teacher network, our method relies on the
> NCP effect, which we demonstrate for the first time for shape matching.
> Thus, the goal of Stage 2 is to *improve* or refine the maps computed in
> Stage 1, which is fundamentally different from the standard
> teacher/student paradigm, where the goal is to *replicate* the results
> of a larger model by a smaller one.
>
> Nevertheless, we agree that a conceptual link does exist and will be
> happy to cite relevant works, and clearly state the difference of our
> setting.
>
> **Q4. Expecting the teacher network to be untrained**
>
> As mentioned in our
> first response, the neural bias section (Section 3.1) was just a
> motivation for our method, which we explained and developed in Section 4
> and Algorithm 1, based on the Neural Correspondence Prior effect that we
> discovered and explained in Section 3.2.
>
> We would like to emphasize that the results in Section 3 are only an
> exploratory study, and that the results in it motivate (as the title of
> the section suggests) the method that we introduce in Section 4. Such a
> separation has been noticed and appreciated by other reviewers such as
> **ZBSs**. We will be happy to further clarify the nature of the relation
> and separation between these sections.
>
> **Q5. Performances of the teacher network on point clouds**
>
> Thank you for
> the question. The performance of the method used for Stage 1 in our
> algorithm (called teacher by the reviewer) is provided in Figure 2 in
> the Supp. Materials, under the label Ours (Stage 1).

---

> ### Author Response · Authors · 2022-08-02
> **Responses to Reviewer 2gYm (Part 2/2)**
>
> **Q6. Our method slightly outperforms the DiffusionNet-based methods**
>
> We would like, first, to clarify that "WSupFMNeT + DiffusionNet" is the
> method we used for Stage 1 of our method (see implementation details in
> Section B of the Supp. Materials), and the maps predicted by this method
> are used for Stage 2. We will make this clear in the revision. We use
> the same method based on unsupervised functional maps in all of our
> experiments to keep our method simple.
>
> Second, we believe that a 2cm improvement in error, or about a 25%
> increase in accuracy, constitutes a significant improvement, especially
> for such difficult problems. Moreover, our improvement is not restricted
> to a particular benchmark or dataset, but is verified broadly on several
> challenging datasets and tasks (point cloud matching (KeyPointNet
> dataset), non-rigid non-isometric shape matching (SMAL and SHREC’20
> datasets), part segmentation (ShapeNet Part dataset) and keypoint
> detection (KeyPointNet dataset)). Finally, our method is fundamentally
> different from previous approaches and, we believe, constitutes a
> conceptual contribution.
>
> **Q7. Figure 4 shows the results of NeuroMorph**
>
> We have shown the results of NeuroMorph because, to the best of our
> knowledge, it is the most recent, state-of-the-art method at the time of
> writing the manuscript. As we clarified in the previous answer,
> "WSupFMNeT + DiffusionNet" is the method we used for Stage 1, and its
> results are shown in Figure 4 under the label "Stage 1".
>
> Nevertheless, we will be happy to add the results of other methods in
> our error figure.
>
> **Novelty is limited, and slight improvement in performance for the
> experiment in Table 2**
>
> We believe that a 25% increase in performance represents a significant
> improvement. In addition, as we shown throughout the main manuscript and
> the Supp. Materials, our methods improve upon the state-of-the-art
> methods on multiple benchmarks, including rigid shape correspondence (up
> to 44% improvement for some classes), shape segmentation transfer (with
> an increase of up to 8% in IOU for some classes), non-rigid shape
> matching, and keypoint detection.
>
> On the other hand, we argue that our method is also conceptually novel.
> As attested by other reviewers, we demonstrate, for the first time, the
> utility of neural prior for shape correspondence, and introduce an
> effect that we call the Neural Correspondence Prior. This effect was not
> observed in prior literature, and we believe that it can lead to
> follow-up works, especially since it is substantiated by significant
> practical utility in our experimental results. We note also that the
> reviewer **4Seq** found the method “novel and interesting”, and the
> reviewer **ZBSs** found the idea “novel and well explained”. We will be
> happy to further clarify our conceptual and practical contribution in
> the final version.
>
> **Exposition**
>
> We thank the reviewer for finding our work presenting
> “simple interesting idea” and the paper “well written”.
>
> We will incorporate *all of the* writing suggestions from the reviewer,
> including making more clarifications, and discussing the link between
> our method and the student/teacher paradigm.
>
> As we have demonstrated throughout the main manuscript, in the
> supplementary materials, and in our response, our paper observes, for
> the first time, the NCP effect, and proposes an algorithm to exploit it.
> Our method achieves state-of-the-art results on multiple benchmarks, as
> evidenced by all the other reviewers. We believe that these points
> address the reviewer’s concern and thus, we kindly ask the reviewer to
> reconsider their rating.

---

### Official Review · Reviewer_41ct · 2022-07-09

**Rating:** 7
**Confidence:** 4
**Soundness:** 3 good
**Presentation:** 3 good
**Contribution:** 4 excellent

**Summary:**

The authors analyze the power of random features from 3D encoders and use it to provide methods for point cloud matching, non-isometric shape matching on meshes, part segmentation transfer and keypoint transfer. It is shown that the random features of several architectures provide a strong prior that automatically resolves matching errors in the early iterations of training. Together with a novel stopping criteria, this leads to a method for refining dense correspondences between shapes.

**Questions:**

- Are many-to-one mappings theoretically allowed / is it just assumed that there is an injective mapping from shape to feature space, as in the requirement of Theorem 1?
- Related to the last question: DiffusionNet is a method that incorporates global information through the global diffusion mechanism. However, other networks, such as PointNet and variants are inherently local, meaning that many-to-one feature mappings will be much more likely, even for isometric shapes. Does the method require a global backend, such as as DiffusionNet (or seemingly PointMLP, not so sure on this one) to work, do the insights in Section 3 only hold for those backends, and in case of a local operator, would an additional consensus method (like [1]) help?
- Would the following statement be correct: The first part of the method finds dense correspondences based on a global feature extractor and fmaps and relies on the near-isometry assumption. Refining the random features during test time is able to resolve errors made due to non-isometric correspondences (in this case, "noise" would not be a reasonable description of such errors). Is this due to the local nature of the random features?
- Many correspondence methods first do local feature matching and then resolve mistakes by introducing global information in a second stage. Here, it seems to be the other way around. Could the authors discuss the implications of this and which advantages/downsides this variant has?
- I would like to know where the proposed method breaks - how mismatching do the shapes have to be? Since the method seem to rely on a global feature extractor backend, how well does it work on partial meshes? Could a similar technique as in the keypoint extractor be applied?
- I think the paper would benefit from an analysis of different initialization schemes, when using random features. In the end, this seems to be the main part of the paper.

[1] Fey et al.: Deep Graph Matching Consensus, ICLR 2020

Minor:
- l. 61: Correspondnece
- l 281: does not relies
- l 285: "point cloud dataset that provide"
- Figure 2 labels are way too small
- Content of the supplementary materials should be explicitly referenced in the main paper (most important: keypoint application).

**Limitations:**

The authors shortly mention some limitations of the method, as it depends on the results of the applied step 1. Discussion of limitations could be improved.
Potential negative societal impact is not discussed but I also see no immediate reason to do so.

**Strengths And Weaknesses:**

Strengths:
- The paper presents a lot of insight into features extracted from 3D operators. The strength of random features was known before (in CNNs) but this is the first time, it was analyzed for 3D architectures on point clouds/meshes
- The described neural correspondence prior in combination with the cycle loss as early-stopping criteria is an interesting and strong mechanism
- There are a lot of state-of-the-art results beaten here (or on par): point cloud matching, shape matching on non-isometric shapes, part segementation transfer, 3D keypoint detection on point clouds
- The paper is well written

Weaknesses:
- The paper raises a lot of questions (see below), I am not sure if I fully understood the effect of the second stage of the network. It would be good if the authors made an additional effort describing the main intuitions.
- Theorem 1 seems to be not that important to me. The crucial parts are the assumption of smooth encoders and injectivity, which are often not given for existing methods (especially local encoders like PointNet). The consequence of smooth encoders and injectivity is pretty straight-forward. It would be more interesting if the authors would additionally show that the used encoders are indeed injective and smooth.
- It seems to me that the term "noise" is used in a hand-wavy fashion in this work. I would appreciate a more clear definition of "noise" and comparisons between different types of noise if they are applied artificially. I am not convinced that the output of stage 1 of the approach should be called "noisy". The errors seem to be more systematic.
- This paper seems to be divided into to main messages, (1) the analysis of random features/neural correspondence prior, and (2) Theorem 1 and its application using the respective networks (DiffusionNet and PointMLP + fmaps). This makes the paper hard to follow in some parts.

Even if the weaknesses list is longer, I think this is a good paper. I liked reading it and it gave me several insights and a few ideas, which is a good sign. In order to make it really great I would encourage the authors to work a bit more on the structure, discussions of main implications and several details that need to be answered, see below.

---

> ### Author Response · Authors · 2022-08-02
> **Responses to Reviewer 41ct (Part 1/3)**
>
> We thank the reviewer for their detailed and thoughtful comments and suggestions. We invite the reviewer to consider the general feedback that we provided to all reviewers in the "High Level Summary" answer. Below are our answers to the reviewers’ questions. Due to the 5000 character limit, and in order to fully address the reviewers’ comments, we have divided our response into three parts.
>
>
> **Q1. Intuition for Stage 2 in our algorithm**
>
> Thank you for your comments
> and suggestions. Our intuition for Stage 2 is threefold.
>
> First, and motivated by the results of Section 3.2 (the NCP effect), we
> showed that neural networks tend to have difficulty overfitting to
> corrupted artifact-laden inputs (neural networks have high noise
> impedance), and in the process of over-fitting an approximate map by a
> neural network, the intermediate maps are of **better quality** than the
> input (L163-166 of the main manuscript), which gives us the idea of
> using artifact-laden maps as supervision for a network, and capitalizing
> on the NCP effect to recover better maps.
>
> Second, and as we mentioned in Section 4.1, when we have a collection of
> approximate maps, their errors tend to be inconsistent, and training a
> network to adapt to them adds an extra layer of regularization, as the
> network cannot adapt to all inconsistent errors on all pairs at once.
>
> Finally, in our formulation of Stage 2, we learn one feature embedding
> per shape, and compute correspondences via nearest neighbor search
> between feature embeddings. This introduces an additional layer of
> regularization, as it makes it difficult for the network to learn a
> feature embedding, which can adapt to errors in the maps between all the
> *pairs of shapes.*
>
> We thank the reviewer for this comment and will be happy to clarify this
> intuition in the final revision.
>
> **Q.3 Is the usage of term “noise” appropriate?**
>
> We thank the reviewer for this question. The use of this term comes from
> our experiment in Section 3.2, where real random noise was used to
> demonstrate the NCP effect, and we have adopted the same terminology
> throughout the paper. With respect to the maps produced by Stage 1, we
> agree with the reviewer that they are not pure noise and that they
> contain some structure. Perhaps a more appropriate term would be
> "approximate" or "artifact-laden" maps. We will make the necessary
> changes in the revision to reflect this idea.
>
> **Q4. Are many-to-one mappings theoretically allowed?**
>
> If the question refers to mappings between shapes, then yes, both in theory and in
> practice many-to-one mappings across shapes are allowed. Note that even
> if the individual feature embeddings are injective, the nearest neighbor
> mapping between them might not be, and the smoothness of the
> correspondence is not hindered by this.
>
> On the other hand, if the question refers to many-to-one feature
> embeddings, then, indeed, as mentioned by the reviewer, we need to
> assume it theoretically, since otherwise the nearest neighbor
> correspondence in feature space is not even well-defined. However, in
> practice, our method works directly without requiring or enforcing
> injectivity of feature embeddings.
>
>
> **Q5. Insights of Section 3 and local operators**
>
> Thank you for the comment.
> Please note that in addition to DiffusionNet and PointMLP, we have
> provided in Table 1 in the Supp. Materials additional results using
> several other backbones: PointNet++, DGCNN and ResidualMLP. The latter
> is a true local backbone, as it applies multiple MLPs to each point
> individually without pooling (PointNet applies pooling at latter
> stages). As can be seen from this table, *all methods* improve the
> results of Stage 1, indicating that the NCP property holds very broadly.
> However, we have observed that using an advanced network such as
> DiffusionNet (which is global by design) tends to yield the best results
> for 3D meshes for example.
>
> **Q7. Advantages/downsides for doing local then global matching?**
>
> Can the reviewer provide some examples of such methods? To the best of our
> knowledge, learning-based shape matching methods all start by finding a
> global map (either by using functional maps or by considering the
> matching problem as a dense semantic segmentation problem), and then
> they try to refine the map locally using techniques such as ZoomOut
> \[4\], PMF \[5\], ICP \[6\], etc.
>
> \[4\] Simone Melzi, Jing Ren, Emanuele Rodolà, Abhishek Sharma, Peter
> Wonka, and Maks Ovsjanikov. 2019. ZoomOut: spectral upsampling for
> efficient shape correspondence. ACM Transactions on Graphics 38, 6
> (2019), 1–14
>
> \[5\] M. Vestner, R. Litman, E. Rodola, A. Bronstein, and D. Cremers.
> Product manifold filter: Non-rigid shape correspondence via kernel
> density estimation in the product space. In Proc. CVPR, pages 6681–6690,
> 2017.
>
> \[6\] Maks Ovsjanikov, Mirela Ben-Chen, Justin Solomon, Adrian Butscher,
> and Leonidas Guibas. Functional Maps: A Flexible Representation of Maps
> Between Shapes. ACM Transactions on Graphics (TOG)

---

> ### Author Response · Authors · 2022-08-02
> **Responses to Reviewer 41ct (Part 2/3)**
>
> **Q2. About Theorem 1**
>
> We thank the reviewer for this comment. As mentioned in our response to
> **ZBSs**, our main motivation behind Theorem 1 is to highlight the fact
> that, given smooth feature embeddings, if we add the injectivity
> condition, the maps extracted using the nearest neighbors in the feature
> space tend to be smooth. As smoothness is a generally desirable
> property, and the frequency bias shown in prior works, such as \[1\]
> suggests that neural networks are biased towards low frequency (and thus
> smooth) functions, we provide this result as a partial explanation for
> the Neural Correspondence Prior, which we have observed, for the first
> time, in our work.
>
> Regarding injectivity, although we agree with the reviewer that such a
> property is not trivial, and might need to be specifically enforced
> (using, for example, invertible networks \[2\]). However, we observe
> that when training a network using contrastive learning, such as NCE
> loss, the latter forces the networks to produce embeddings that are
> unique for each point, in order to minimize the NCE loss, which can be
> considered a form of injectivity.
>
> Last but not least, we agree with the reviewer that the conditions of
> the theorem are not trivial. Furthermore, as mentioned in our response
> to **ZBSs**, we do not consider this theorem as a key contribution of
> our work. We would therefore be happy to paraphrase it as simply a
> high-level motivation for our approach, if the reviewers deem
> appropriate.
>
> Finally, we include here the results of the experiments suggested by
> **ZBSs**, and also by the reviewer, which is checking the smoothness and
> injectivity of a randomly initialized network. For this, we performed
> the following experiment. We first computed the embeddings produced by a
> randomly initialized DiffusionNet network for the 20 test shapes in the
> FAUST remeshed dataset.
>
> To evaluate the smoothness of the embedding, we computed the standard
> Dirichlet energy of both the embedding produced by the network and the
> original embedding of the shape in $\mathbb{R}^3$ (the 3D coordinates of
> the shape’s vertices), using the following formula:
> $$E_{Dirichlet}(G) = \frac{1}{n} \sum_{i=1}^n \frac{G_i^{\top} W G_i}{G_i^{\top} A G_i},$$
> where $G$ is the embedding in question of dimension $n$, while $W$ and
> $A$ are, respectively, the standard stiffness and mass matrices,
> computed using the classical cotangent discretization scheme of the
> Laplace-Beltrami operator \[3\]. We computed the Dirichlet energy over
> the whole test set and found it equal to **47.2** (the average) for
> coordinate functions (original shape embeddings in $\mathbb{R}^3$),
> while it is equal to **13.1** for embeddings produced by the network.
> This shows that the latter are smoother than embeddings in the original
> space.
>
> For injectivity, we computed for each point of the embedding, the
> distance to its nearest neighbor, and took *the minimum* across all
> points of a shape. To make the comparison between the original embedding
> in $\mathbb{R}^3$ and the embedding produced by the network fair, we
> normalized both embeddings to the unit sphere. We find that on average,
> the minimum distance between points in the original 3D space is
> **0.0004**, while for the feature embedding, given by the network it is
> equal to **0.0015**. This shows that the network is injective, and that
> distances between points are larger than in the original domain.
>
> We will be happy to incorporate this result in the final version.
>
> \[1\] Rahaman, Nasim, et al. "On the spectral bias of neural networks."
> International Conference on Machine Learning. PMLR, 2019.\]
>
> \[2\] Behrmann, J., Grathwohl, W., Chen, R. T., Duvenaud, D., and
> Jacobsen, J.-H. Invertible residual networks. In International
> Conference on Machine Learning, pp. 573–582, 2019.
>
> \[3\] Ulrich Pinkall and Konrad Polthier. Computing discrete minimal
> surfaces and their conjugates. Experimental mathematics, 2(1):15–36,
> 1993
>
> **Q8. Where the proposed method breaks?**
>
> We observed that our method fails
> either when the maps in Stage 1 are total noise and have no structure,
> or when the maps are symmetrically inverted (e.g., mapping left to right
> on human or animal shapes), in which case it is difficult to recover a
> ground truth map from this initialization, since structurally such maps
> are identical to the ground truth.
>
> Regarding partial shapes, we thank the reviewer for this suggestion, and
> we think this is an interesting direction for future research. We
> believe that our method will still work in this setting, but additional
> attention must be paid to partiality, either by using a partiality
> criterion such as the one used in our keypoint detection experiment, or
> by using a special network for detecing the overlap region, such as the
> one introduced by DPFM \[7\].
>
> \[7\] Souhaib Attaiki, Gautam Pai, and Maks Ovsjanikov. DPFM: Deep
> partial functional maps. In 2021 International Conference on 3D Vision
> (3DV). IEEE, December 2021.

---

> ### Author Response · Authors · 2022-08-02
> **Responses to Reviewer 41ct (Part 3/3)**
>
> **Q6. Is the statement provided by the reviewer correct?**
>
> The first part of
> the statement is not correct. First, we do not necessarily need a
> feature extractor for Stage 1, any off-the-shelf unsupervised shape
> matching method could work.
>
> Please see Table 2 in the Supp. Materials where we included a
> demonstration of this: we used 4 different methods for Stage 1, one of
> which is Smooth Shells, which is an axiomatic and not a learning-based
> method.
>
> Second, we do not always rely on the quasi-isometry assumption in Stage
> 1, as for example in the Rigid Shapes experiment (Section 5.1.1 of the
> main manuscript), where the network is trained only with the bijectivity
> loss (see Section B of the Supp. Materials for more details on the
> implementation), and no near-isometry assumption is made.
>
> For the second part of the statement, our Stage 2 does indeed solve for
> errors related to non-isometric matches, as well as other types of
> errors, introduced by artifacts in Stage 1. Figure 1 in the main
> manuscript supports this statement, where the network is able to denoise
> maps that are corrupted by purely random noise, up to 75% (i.e., 75% of
> the entries of the map are noise).
>
> **Q9. Analysis of different initialization schemes**
>
> We are not sure to
> fully understand the reviewer’s question. If by initialization schemes
> the reviewer means the method used in Stage 1 to get the
> "artifact-laden" maps, we did such a study in Table 2 of the Supp.
> Materials, where we tested 4 different initialization methods (Smooth
> Shells, Unsup GeomFMaps, Deep Shells, and NeuroMorph), and showed that
> the method always works, and does not depend on the choice of Stage 1.
>
> If by random initialization, the reviewer means the different
> initialization schemes of a neural network (such as normal
> initialization \[8\], Xavier initialization \[9\], Kaiming
> initialization \[10\], etc.), we want to emphasize that we used the
> default initialization in Pytorch for all our networks (which is a
> combination of Xavier initialization, and Kaiming initialization).
>
> Furthermore, please note that our Algorithm 1, stated in Section 4.1.
> and used in all of our results, does not use features produced by a
> randomly initialized network (described in Section 3.1), but rather the
> output of an unsupervised correspondence method that we refine in our
> Stage 2.
>
> \[8\] Y. LeCun, L. Bottou, G. B. Orr, and K. R. Muller, Efficient
> backprop, In G. B. Orr and K. R. Muller, editors, Neural Networks:
> Tricks of the Trade, pp. 9-50, Springer, 1998
>
> \[9\] Glorot, Xavier, and Yoshua Bengio. "Understanding the difficulty
> of training deep feedforward neural networks." Proceedings of the
> thirteenth international conference on artificial intelligence and
> statistics. JMLR Workshop and Conference Proceedings, 2010.
>
> \[10\] Kaiming He, Xiangyu Zhang, Shaoqing Ren, and Jian Sun. Delving
> deep into rectifiers: Surpassing human-level performance on ImageNet
> classification. In Proceedings of the IEEE international conference on
> computer vision, pages 1026–1034, 2015.
>
> **Limitation and Negative Impact**
>
> We will be happy to expand upon the
> limitations section. As mentioned above (in the answer to Q8), our method tends to break when
> the maps in Stage 1 either have no structure, or when they are
> symmetrically inverted. Moreover, since our method is general, it might
> not be the best choice, e.g., for near isometric categories, where,
> specialized methods could perform better. In contrast, we target
> difficult non-rigid non-isometric datasets, where existing methods tend
> to fail.
>
> Please note that we provided a section on societal impact in Section G
> of the Supp. Materials, however, as the reviewer said, we could not see
> any immediate negative impact.
>
> **Exposition** We thank the reviewer for finding our paper “good,
> interesting and well written”.
>
> We will be happy to incorporate writing suggestions from the reviewer,
> including more clarifications, rearranging the sections, and correcting
> the spelling mistakes.

---

> > ### Comment · Reviewer_41ct · 2022-08-09
> > **Answer to authors**
> >
> > I thank the authors for the additional clarifications. They were helpful for further understanding.
> >
> > Q2: I think the additional evaluation of smoothness and injectivity is an important addition. They should be added to the paper. I would be also interested in a comparison between different networks.
> >
> > Q7: I think we are talking about the same methods. E.g. functional maps require descriptors in order to find the map C in the spectral domain. Initial descriptors are often found using local methods. Then, a global correspondence that aligns local descriptors is found in the spectral domain. But I agree that this distinction is not clear and the border between local and global is not clearly defined, as some descriptors are found using global methods, as it is the case here, depending on the backbone.
> >
> > Q9 was about the network initialization for the first part of the paper (neural bias for pointwise features). I would imagine that the network initialization scheme has an impact on the result (given that the point embeddings are generated purely based on this random initialization) and still think that an analysis of different schemes would make the paper better.
> >
> > All in all, (1) the rebuttal did improve clarity of the work and (2) an important experiment was added that supports the assumptions of Theorem 1. I increased my score to 7 as I think this paper should get accepted.

---

> > > ### Author Response · Authors · 2022-08-09
> > > **Response to the Reviewer**
> > >
> > > We thank the reviewer for their response and constructive feedback, as well as for considering our rebuttal and increasing the score for our paper. We will make sure to include the evaluation of smoothness and injectivity, as suggested by the reviewer. We will also make sure to include an analysis of the different initialization schemes of the random network in the final version.

---

### Official Review · Reviewer_ZBSs · 2022-07-11

**Rating:** 7
**Confidence:** 4
**Soundness:** 3 good
**Presentation:** 3 good
**Contribution:** 3 good

**Summary:**

This paper applies the idea of deep prior to predict 3D correspondences. It does so using two ideas, which are demonstrated experimentally in the first part of the paper: first, it shows that randomly intialized diffusionNet features actually provide better results than spectral and SHOT   descriptors, and combined with fonctional maps even improve over learned metods; second, it shows that when training a network to predict features to fit noisy correspondences, the quality of correspondences/features obtained at the beginning of training are actually better than the noisy target. This naturally leads to a method to predict correspondences, obtaining features from a randomly initialized network, then training a second one to predict them and stoping early (using a cycle consistency loss to decide when)

**Questions:**

See weaknesses (in particular FAUST evaluation)

**Limitations:**

yes

**Strengths And Weaknesses:**

I liked this paper, the idea is simple, well explained and well demonstrated. To the best of my knowledge using neural prior for shape correspondence is novel, if this is indeed the case I think the paper should clearly be accepted.

As for weaknesses:
- Why aren't correspondences evaluated on FAUST? I would really want to see this evaluation, which would also ease comparison with other methods.
- smaller weaknesses:
* I liked the separation of section 3 and 4, but then section 4 points several times to section 5 for important details about the method, I think this makes the reading more complicated than needed, please move all details about the method to section 4.
* I am not a fan of theorem 1. It is actually not related directly to the fact that the mapping are neural nets, it would need a demonstration that this is the general case (i.e. with random weights) for the neural net architecture considered, and both formulation and demonstration in the supmat are very handwavy (for example the theorem is clearly false for d<3, which doesn't appear anywhere). I would thus suggest to completely remove it, or at least not call it a theorem for which I would like a much more rigorous demonstration
* too many unnecessary abbreviations make the text harder to read (FR, SR, NCP, NCP-UN, UM, RN not counting the ones for previous methods ), I would remove at least 50% of them and be sure to regularly remind the reader of the meaning of the ones kept (none, or maybe only NCP would be great)
* fig 2 is not readable in the printed version

---

> ### Author Response · Authors · 2022-08-02
> **Responses to Reviewer ZBSs (Part 1/2)**
>
> We thank the reviewer for their detailed and thoughtful comments and suggestions. We invite the reviewer to consider the general feedback that we provided to all reviewers in the "High Level Summary" answer. Below are our answers to the reviewers’ questions. Due to the 5000 character limit, and in order to fully address the reviewers’ comments, we have divided our response into two parts.
>
>
> **Q1. Correspondences evaluated on FAUST**
>
> Thank you for the suggestion. In
> our work, we focus on difficult non-rigid non-isometric datasets, where
> existing methods tend to fail. This is because, on near-isometric
> datasets such as FAUST or SCAPE, current unsupervised methods can
> exploit the assumption of near-isometry and achieve good results.
>
> Nevertheless, as suggested, below we include a comparison of our method
> on the FAUST remeshed and SCAPE remeshed datasets, using the same
> train/test split used in all previous works. The notation X on Y means
> the method is trained on X and tested on Y.
>
> As input for our Stage 2, we use WSupFMNet + DiffusionNet \[1\] (first
> line of the table), which is the same method used in the main manuscript
> for all our experiments. For our method (Stage 2), we use the same
> implementation as the one used for the SMAL dataset, described in
> Section 5.1.2 of the main manuscript.
>
> As shown in this table, even on a near-isometric dataset, our Stage 2
> improves upon the initial maps in all categories, by 18.4% on average.
> Nevertheless, we remark that in such settings it is more advantageous to
> use specialized methods, such as ZoomOut \[2\], that directly exploit
> the near-isometry assumption.
>
> We will be happy to include this result and discussion in the final
> version.
>
> | Method                             | FAUST on FAUST | SCAPE on SCAPE | FAUST on SCAPE | SCAPE on FAUST |
> |:-----------------------------------|:--------------:|:--------------:|:--------------:|:--------------:|
> | WSupFMNet + DiffusionNet           |       3.8      |       4.4      |       4.8      |       3.6      |
> | WSupFMNet + DiffusionNet + ZoomOut |       1.9      |       2.6      |       2.7      |       1.9      |
> | NCP (ours)                         |       3.0      |       3.5      |       4.2      |       2.9      |
> | NCP (ours) + ZoomOut               |       1.9      |       2.4      |       2.6      |       1.9      |
>
> ---
>
> \[1\] Nicholas Sharp, Souhaib Attaiki, Keenan Crane, and Maks
> Ovsjanikov. Diffusionnet: Discretization agnostic learning on surfaces.
> ACM Trans. Graph., 01(1), 2022.
>
> \[2\] Simone Melzi, Jing Ren, Emanuele Rodolà, Abhishek Sharma, Peter
> Wonka, and Maks Ovsjanikov. 2019. ZoomOut: spectral upsampling for
> efficient shape correspondence. ACM Transactions on Graphics 38, 6
> (2019), 1–14

---

> ### Author Response · Authors · 2022-08-02
> **Responses to Reviewer ZBSs (Part 2/2)**
>
> **Q2. Utility of Theorem 1**
>
> We thank the reviewer for this comment. The
> motivation behind Theorem 1 is to highlight the fact that, given smooth
> feature embeddings, if we add the injectivity condition, the maps
> extracted using the nearest neighbors in the feature space tend to be
> smooth. As smoothness is a generally desirable property for maps, and
> the frequency bias shown in prior works, such as \[3\] suggests that
> neural networks are biased towards low frequency (and thus smooth)
> functions, we provide this result as a partial explanation for the
> Neural Correspondence Prior, which we have observed, for the first time,
> in our work.
>
> Nevertheless, we agree with the reviewer that this theorem is not
> related directly to the fact that the mapping is given by a neural
> network. Furthermore, we do not consider this as a key theoretical
> contribution of our work. We would therefore be happy to paraphrase it
> as simply a high-level motivation for our approach, as suggested by the
> reviewer.
>
> Remark, however, that we believe the result still does hold for d\<3.
> For d=1, there are no smooth injective embeddings of 2d manifolds, and
> thus the theorem does not apply (as it requires such an embedding by
> assumption). For 2d, *if* the feature embeddings are smooth and
> injective, then the nearest neighbor map will generically be smooth up
> to sets of measure zero. We will be happy to provide more details on the
> argument, if necessary.
>
> Furthermore, following the reviewer’s suggestion, we evaluated the
> smoothness and injectivity of a randomly initialized network.
> Specifically, we first computed the embeddings produced by a randomly
> initialized DiffusionNet network for the 20 test shapes in the FAUST
> remeshed dataset.
>
> To evaluate the smoothness of the embedding, we computed the standard
> Dirichlet energy of both the embedding produced by the network and the
> original embedding of the shape in $\mathbb{R}^3$ (the 3D coordinates of
> the shape’s vertices), using the following formula:
> $$E_{Dirichlet}(G) = \frac{1}{n} \sum_{i=1}^n \frac{G_i^{\top} W G_i}{G_i^{\top} A G_i},$$
> where $G$ is the embedding in question of dimension $n$, while $W$ and
> $A$ are, respectively, the standard stiffness and mass matrices,
> computed using the classical cotangent discretization scheme of the
> Laplace-Beltrami operator \[4\]. We computed the Dirichlet energy over
> the whole test set and found it equal to **47.2** (the average) for
> coordinate functions (original shape embeddings in $\mathbb{R}^3$),
> while it is equal to **13.1** for embeddings produced by the network.
> This shows that the latter are smoother than embeddings in the original
> space.
>
> For injectivity, we computed for each point of the embedding, the
> distance to its nearest neighbor, and took *the minimum* across all
> points of a shape. To make the comparison between the embedding in
> $\mathbb{R}^3$ and the embedding produced by the network fair, we
> normalized both embeddings to the unit sphere. We find that on average,
> the minimum distance between points in the original 3D space is
> **0.0004**, while for the feature embedding, given by the network it is
> equal to **0.0015**. This shows that the network is injective, and that
> distances between points are larger than in the original domain.
>
> We will be happy to incorporate this result in the final version.
>
> \[3\] Rahaman, Nasim, et al. "On the spectral bias of neural networks."
> International Conference on Machine Learning. PMLR, 2019.\]
>
> \[4\] Ulrich Pinkall and Konrad Polthier. Computing discrete minimal
> surfaces and their conjugates. Experimental mathematics, 2(1):15–36,
> 1993
>
> **Exposition**
>
> We thank the reviewer for finding our paper “simple, well
> explained and well demonstrated”. We also thank the reviewer for
> acknowledging that our work is the first to demonstrate the utility of
> the neural prior for shape correspondence.
>
> We will incorporate *all of the writing suggestions* from the reviewer,
> including more clarifications, fixing the font size and removing the
> unnecessary abbreviations.

---

### Official Review · Reviewer_4Seq · 2022-07-12

**Rating:** 6
**Confidence:** 4
**Soundness:** 3 good
**Presentation:** 3 good
**Contribution:** 3 good

**Summary:**

This paper proposed an unsupervised shape matching method, where the core idea is to first generate noisy maps from existing unsupervised shape matching methods, and refine them by learning pointwise features from the noisy maps, and build a new map by matching those features. The new map is empirically more accurate than the input noisy maps. Experiments are conducted to support the claim.


**Questions:**

1. Is it possible if one repeats the process of learning features from a map and matching the features to produce a map, i.e., steps 2-5, would one get further improvement?
2. What is the main difference between 4.1 NCP and 4.2 test time denoising? In 4.2 do you train an extra network beyond that already trained in 4.1 NCP? Is the purpose of 4.2 to propose an early stopping criterion for NCP?


**Strengths And Weaknesses:**

The reviewer finds the work novel and interesting. The main concerns are as follows:

1. How is the loss in line 260 differentiable with respect to the network parameters? Since it appears that Pi_MN lies in the set of permutations, which is a discrete set.
2. In section 3.2 it says early stopping is needed for training the random-initialized network to learn features. However, it appears that early stopping criterion is only proposed in 4.2 but not 4.1. Is it true that 4.2 is not an optional step of denoising, but actually an important step for early stopping?


Minor issues on presentation:
3. The notations of Theorem 1 (lines 201-204) look a bit confusing. Does N(M)_x mean the mapping N restricted on M applied on x \in M?
If so, I wonder if N(M)_x a standard notation, or if N|_M(x) is more standard? Since in the former, N(M) could mean the image of M under N.
If not, what does this notation mean?
4. The fonts in the figures could be improved, in particular, they are way too small compared to the body text, and it seems that there is space.

---

> ### Author Response · Authors · 2022-08-02
> **Responses to Reviewer 4Seq**
>
> We thank the reviewer for their detailed and thoughtful comments and suggestions. We invite the reviewer to consider the general feedback that we provided to all reviewers in the "High Level Summary" answer. Below are our answers to the reviewers’ questions.
>
>
>
> **Q1. Is the loss in line 260 differentiable?**
>
> Thank you for your comment. In our implementation, $\Pi_{MN}$ is a soft matrix computed using Eq. (5) in the Supp. Materials. This makes the whole operation differentiable. We will clarify this in the revision.
>
> **Q2. Is early stopping an important step in 4.2?**
>
> To clarify, as shown in
> Figure 1 of the main manuscript, when using a single pair of shapes (as
> in Section 4.2), if no early stopping is used, the network will
> eventually overfit to the noisy map, making early stopping necessary.
>
> However, when using a collection (the setting in Section 4.1), as
> mentioned on L187-190 of the main manuscript, the collection provides an
> additional layer of regularization, making early stopping unnecessary.
> Since *all* of our results, except those explicitly mentioned in Table
> 2, are produced using the approach described in Section 4.1, we do not
> rely on early stopping in practice.
>
> In Table 2 we presented results with both methods (Ours, which uses the
> collection) and (Ours, Test-Time Optimization, using the method
> described in Section 4.2, applied on each pair independently). As can be
> seen from this table, the results when using the collection are better.
> Furthermore, this method is also faster since it does not require
> test-time optimization on each shape individually.
>
> In summary, early stopping is only required when considering individual
> shape pairs. In our experimental results, we are given access to a
> collection of training shapes, and thus we use the method in Section
> 4.1, which makes early stopping unnecessary. We will be happy to make
> this more explicit.
>
> **Q3. About the notation of Theorem 1**
>
> We use $N(M)_x$ to denote the image
> of the mapping $N(M)$ at some point $x$ ($N(M)$ is the image of the
> network $N$ applied to the surface $M$). An alternative, possibly more
> standard notation, would be to use $N(M)|_x$, i.e., the mapping $N(M)$,
> restricted to $x$. We will be happy to make this change.
>
> **Q4. Repeating the second stage multiple times, i.e steps 2-5 in the
> algorithm**
>
> Yes, it is possible to repeat Stage 2 several times, and we
> observed that there is a slight improvement in some cases, but most
> often it stagnates after the first iteration. We experimented with this
> and chose to keep the method simple and use only one iteration (thus,
> avoiding an additional tunable hyperparameter). We will be happy to
> provide illustrations of this phenomenon.
>
> **Q5. The main difference between 4.1 NCP and 4.2 test time denoising**
>
> The difference between the methods in Section 4.1 and 4.2 lies in the
> amount of training data available. The method proposed in Section 4.2 is
> used when we only have a single pair of shapes and an approximate map
> between them as input. In this case, we train a network to overfit to
> this input map and use early stopping to achieve improvement in
> accuracy.
>
> The method in Section 4.1 is used when we have a collection of shapes at
> training time (as is the case in *all of our experiments*, except the
> one marked “test-time optimization” in Table 2 of our main manuscript).
> In this case, in Stage 1, we use an unsupervised method to get an
> initial set of maps. In Stage 2, we then use these approximate maps as
> supervisory signal, as described in Algorithm 1. We advocate this method
> in case of collections because 1) it avoids early stopping, 2) it leads
> to more accurate results, and 3) it is faster as no test time
> optimization is necessary, and the network, once trained, can be used
> directly to predict high quality maps on unseen shape pairs.
>
> **Exposition**
>
> We thank the reviewer for the suggestions regarding the
> exposition. We will incorporate *all of the* writing suggestions from
> the reviewer, including making more clarifications, and enlarging the
> font size in the figures. We also thank the reviewer for finding our
> work “novel and interesting”.

---

### Author Response · Authors · 2022-08-02
**High Level Summary**

We thank the reviewers for their constructive comments. We find the
suggestions to be very helpful for improving the quality of our work,
making it clearer and more convincing.

Before responding to individual concerns, we stress the following
contributions of our work:

**(1)** We show, for the first time, that when trained using
approximate, artifact-laden maps for supervision, the features learned
by neural networks lead to correspondences that are more coherent and
are of higher quality than the input, including on difficult non-rigid
non-isometric shape pairs. We call this effect Neural Correspondence
Prior \[**4Seq**, **41ct**\]. As remarked by \[**ZBSs**\] our work
demonstrates, for the first time, “neural prior for shape
correspondence”.

**(2)** We demonstrate that an untrained recent state-of-the-art neural
network for 3D shapes can produce pointwise features that are on par
with complex axiomatic local descriptors \[**ZBSs**, **41ct**,
**2gYm**\]

**(3)** Based on these insights, we develop a two-stage algorithm for
unsupervised 3D non-rigid shape matching that achieves SOTA results in a
broad range of difficult matching scenarios including non-isometric and
point cloud matching, as well as other tasks such as part segmentation
transfer and 3D keypoint detection \[**ZBSs**, **41ct**\].

We believe that all of the suggested changes can be easily done within a
minor revision and we will make sure to address all of the comments and
concerns in the final version. We will also release our code and data
for full reproducibility of our results and to facilitate future work in
this area.

---

### Meta-Review · Area_Chair_ppZb · 2022-08-23

**Recommendation:** Accept
**Confidence:** Certain

**Metareview:**

This paper received mixed scores, with three reviewer recommending acceptance and one rejection. The reviewers appreciated the simplicity and effectiveness of the method, but nonetheless raised many questions about the method, requesting the authors to clarify several points. The authors' feedback addressed most of these questions. During the discussion, 2gYm, the most negative reviewer, mentioned that they found the contributions interesting for the shape matching community but not significant enough to be published in NeurIPS. Considering that three reviewers found this work sufficiently interesting to recommend acceptance, the AC deems this to be a secondary concern. The AC nonetheless strongly encourages the authors to revise their paper based on their feedback for the camera-ready version.

**Award:**

No

---

### Decision · Program_Chairs · 2022-09-14

Accept